# scGCN is a graph convolutional networks algorithm for knowledge transfer in single cell omics

Qianqian Song[1,2], Jing Su [3,4✉] & Wei Zhang [1,2✉]

Single-cell omics is the fastest-growing type of genomics data in the literature and public genomics repositories. Leveraging the growing repository of labeled datasets and transferring labels from existing datasets to newly generated datasets will empower the exploration of single-cell omics data. However, the current label transfer methods have limited performance, largely due to the intrinsic heterogeneity among cell populations and extrinsic differences between datasets. Here, we present a robust graph artificial intelligence model, single-cell Graph Convolutional Network (scGCN), to achieve effective knowledge transfer across disparate datasets. Through benchmarking with other label transfer methods on a total of 30 single cell omics datasets, scGCN consistently demonstrates superior accuracy on leveraging cells from different tissues, platforms, and species, as well as cells profiled at different molecular layers. scGCN is implemented as an integrated workflow as a python software, which is available at https://github.com/QSong-github/scGCN.

[1] Center for Cancer Genomics and Precision Oncology, Wake Forest Baptist Comprehensive Cancer Center,  Wake Forest Baptist Medical Center, Winston Salem, NC, USA. [2] Department of Cancer Biology, Wake Forest School of Medicine, Winston Salem, NC, USA. [3] Department of Biostatistics and Health Data Science, Indiana University School of Medicine, Indianapolis, IN, USA. [4] Section on Gerontology and Geriatric Medicine, Department of Internal Medicine, Wake Forest School of Medicine, Winston-Salem, NC, USA. ✉email: su1@iu.edu; wezhang@wakehealth.edu

Single-cell omics technologies are increasingly used in biomedical research to provide high resolution insights into the complex cellular ecosystem and underlying molecular interconnectedness[1–3]. Leading this wave of omics is single-cell RNA sequencing (scRNA-seq), which allows measurement of the transcriptome in thousands of single cells from multiple biological samples under various conditions[4–8]. Other single-cell-based assays, include Single-cell Assay for Transposase-Accessible Chromatin using sequencing (scATAC-seq), profile cellular heterogeneity at the epigenetic level[9–11], which further elucidates transcriptional regulators[8,12]. These technological developments allow profiling of multiple molecular layers at single-cell resolution and assaying cells from multiple samples under different conditions.

The rapid advances in single-cell technologies have led to remarkable growth of single cell omics data. As more and more single-cell datasets become available, there is an urgent need to leverage existing and newly generated data in a reliable and reproducible way, learning from the established single-cell data with well-defined labels as reference, and transferring labels to newly generated datasets to assign cell-level annotations[10,11]. However, existing datasets and newly generated datasets are often collected from different tissues and species[13,14], under various experimental conditions, generated by different platforms[15,16], and in different omics types[17]. Thus a reliable and accurate knowledge transfer method must overcome the following challenges: (1) the unique technical issues of single-cell data (e.g., dropouts and dispersion)[18–21]; (2) batch effects that arise from different operators, experimental protocols[16], and technical variation (e.g., mRNA quality, pre-amplification efficiency, technical settings during data generation)[22–24]; and (3) intrinsic biological variances associated with different tissues, species, and molecular layers such as RNA-seq and ATAC-seq.

To address these challenges in transferring labels across different datasets, several methods have been developed. The most commonly used are Seurat v3[25,26] and the recently reported Conos[27], scmap[28], and CHETAH[29]. Seurat is a well-established, widely used toolkit for single cell genomics[25,26]. Recently an anchor-based label transfer method across substantially different single-cell samples has been proposed in Seurat v3[26]. Conos generates a joint graph representation by pairwise alignments of samples, to propagate labels from one sample to another. scmap learns cell types by measuring the maximum similarity between the reference dataset with well-annotated cells and unknown datasets[30]. Guided by the reference data, CHETAH identifies a classification tree for a top-to-bottom classification in unannotated data. Whereas these methods are valuable in different settings, they exhibit limited capability and performance, partially due to the fact that they only extract shared information from individual cells but ignore higher-order relations between cells. Such topological cell relation can be well captured by the Graph Convolutional Networks (GCN)[31]. Recently, GCN and its related methods have been successfully applied in single cells and in diseases[32–36], showing that inclusion of GCN enables learning of high-order representation and topological relations of cells that improve performance.

Here, we present a graph-based artificial intelligence model, termed single-cell Graph Convolutional Network (scGCN). We provide evidence that scGCN allows reliable and reproducible integration of single-cell datasets and transferring labels across studies. Thus, knowledge learned from well-characterized datasets in previous studies can be transferred to and provides insights in studies. Using a wide range of different single-cell omics datasets, including data from different tissues, species, sequencing platforms, and molecular layers (such as RNA-seq and ATAC-seq), we demonstrate that scGCN outperforms other methods in accuracy and reproducibility. In addition, we provide the implementation software of scGCN, which is compatible with various single-cell datasets for accurate cell type identification.

## Results

**Overview of the scGCN.** Knowledge learned from existing single-cell datasets is often represented as cell labels. Examples of cell labels are cell type, developmental state, activation status, cellular functionality, and signaling pattern. The scGCN approach leverages well-characterized single-cell data as reference to infer such cellular-level knowledge in query dataset, i.e., label transfer, through semi-supervised learning (Fig. 1). First, scGCN learns a sparse and hybrid graph of both inter-dataset and intra-dataset cell mappings using mutual nearest neighbors of canonical

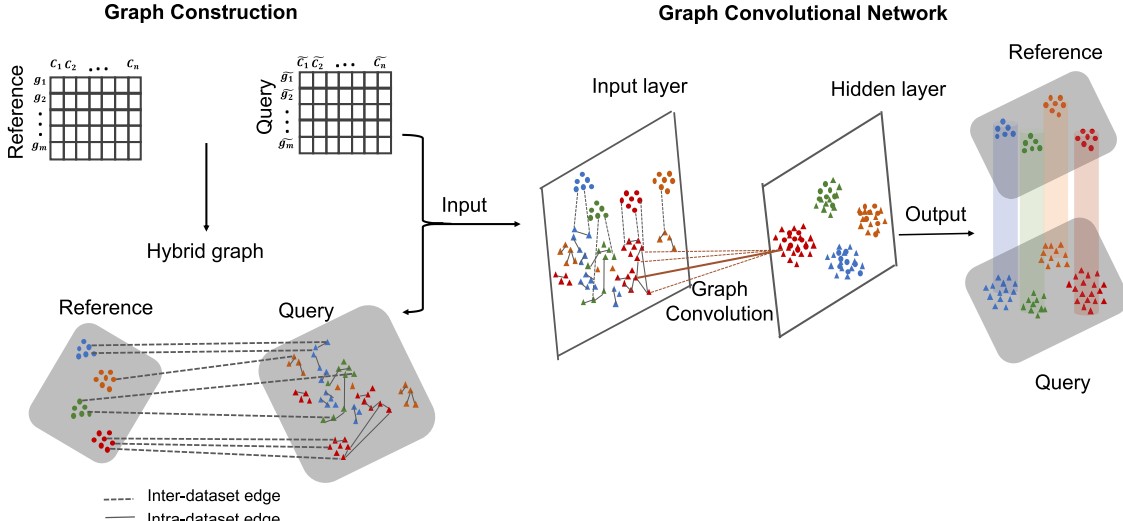

**Fig. 1 Schematic overview of scGCN for transferring labels from reference data to query data.** scGCN first learns a hybrid graph of both inter-dataset (dash line) and intra-dataset (solid line) cell mappings. In the hybrid graph, round solid-colored dots represent cells of the reference dataset, while the triangles represent cells from the query dataset. Based on the hybrid graph, semi-supervised GCN is used to project cells of both reference and query datasets onto hidden layer so that cells with the same labels present in the same subpopulation. Thus, cell labels in query data are predicted and learned from reference data.

correlation vectors that project different datasets onto a correlated low-dimensional space, which enables the identification and propagation of shared information between reference and query data. Subsequently, based on the constructed graph, semi-supervised GCN is used to project cells of both reference and new datasets onto the same latent space so that cells with the same labels present in the same population. Consequently, cell labels in query data are predicted and learned from reference data. Details are described in "Methods" section.

**Performance of label transfer within datasets.** To evaluate the performance of scGCN, we benchmark it against other methods, including Seurat v3[26], Conos[27], scmap[28], and CHETAH[29]. For quantitative benchmarks, we first use ten scRNA-seq datasets that vary in cell numbers, tissues, species, and sequencing technologies. These datasets represent different scenarios and challenges in cell label transfer. For each dataset, we randomly select 50% of its cells as the reference data and apply the above methods to learn the labels of the other 50% of cells as the query data.

We evaluate the performance of each method using the accuracy score (Acc), which is defined as the proportion of correctly predicted cells among all cells in the query data. Our results show that scGCN consistently demonstrates better performance than other methods across datasets and achieves the highest accuracy (mean Acc = 91%, Fig. 2a), which is significantly higher than other methods (Seurat v3: 87%; Conos: 83%; scmap: 82%; CHETAH: 77%) ($P$ value = 0.0019, 0.0058, 0.0039, and 0.002, respectively). scGCN demonstrates particularly higher accuracy relative to these methods in some datasets. For example, Seurat v3 shows a relatively lower accuracy score in GSE99254 dataset (Seurat v3: 77%; scGCN: 85%), while Conos performs poorly on the GSE108989 dataset (Conos: 53%; scGCN: 86%), and CHETAH performs poorly on the SRP073767 dataset (CHETAH: 62%; scGCN: 90%). Therefore, scGCN achieves the best performance in transferring labels accurately in the ten benchmarking datasets.

To highlight the comparison regarding specific cell types, we use the SRP073767 dataset as an example, which has two subtypes of T cells (CD4+ T helper2 cells, CD4+/CD25 Treg cells) that every benchmarked method cannot distinguish accurately whereas scGCN performs relatively better (Fig. 2b and Supplementary Fig. 1a). In addition, Seurat v3 and Conos cannot discriminate between CD4+ T helper2 cells and CD4+/CD45RA+/CD2− Naive T cells, nor the CD8+/CD45RA+ Naive Cytotoxic cells and CD8+ Cytotoxic T cells. scmap and CHETAH also inaccurately assigns CD4+ T cells to CD8+ T cells, and has low accuracy for monocyte classification and unassigned cells. scGCN, in contrast, performs better than other methods in discerning these similar cell types.

**Transfer labels across datasets of different platforms.** As emerging single cell data are generated by different experimental platforms, we test whether scGCN can be used to accurately transfer labels across datasets from different platforms. Here we include 12 paired reference-query datasets. Each pair of reference-query datasets is profiled using different scRNA-seq technologies.

Similarly, we use the accuracy score to evaluate the performance of each method. Based on the 12 reference-query datasets, the accuracy score of scGCN (mean Acc = 87%, Fig. 3a) is consistently higher than Seurat v3 (mean Acc = 82.2%) and Conos (mean Acc = 82.3%), and also significantly better than scmap (mean Acc = 66%; $P$ value = 0.001433) and CHETAH (mean Acc = 58%, $P$ value = 2.219e−05). Specifically, when querying the PBMC Smart-seq2 data from the reference of PBMC Dropseq data, both Seurat v3

(Acc = 68%) and scmap (Acc = 60%) show relatively lower accuracy scores than scGCN (Acc = 77%). Similarly, Conos (Acc = 71%) and CHETAH (Acc = 32%) present lower accuracy than scGCN (Acc = 76%) when mapping MCA Smart-seq2 from MCA 10× data. Notably, when annotating the PBMC 10× V3 data using the PBMC Cel-seq data as reference, scGCN is distinctively superior to other methods. Although these two datasets are generated using different platforms, Uniform Manifold Approximation and Projection (UMAP)[37] reveals highly consistent transferred labels by scGCN for the cell populations of the PBMC 10× V3 query data (Fig. 3b). The Sankey diagram shows that the reference dataset is much smaller than the query dataset (Fig. 3c), suggesting the effectiveness of scGCN even with a small reference dataset.

We next use heatmap to depict the accuracy of each cell type, including B cells, CD4+ T cells, and cytotoxic T cells identified by different methods (Fig. 3d and Supplementary Fig. 1b). Seurat v3 and CHETAH incorrectly assign most CD4+ T cells and natural killer cells to cytotoxic T cells. Some megakaryocytes and CD16+ monocytes are assigned to other cell types by Conos and scmap. The closely related cell types such as natural killer cells and T cells, as well as CD14+ monocytes and CD16+ monocytes, are correctly discriminated by scGCN.

**Transfer labels across datasets of different species.** We next evaluate the performance of scGCN across datasets from different species. We apply all benchmarking methods to four paired reference-query datasets. For each paired dataset, one consists of cells from mouse and the other from human tissues.

We first apply all five methods to identify the labels of query data in the four pairs of datasets. Then we compare the visualization of cells using the aggregated reference-query data by different methods (Fig. 4a). Because CHETAH does not provide the aggregated data, we omit this method in the following comparisons of cell visualization. Due to the inherent noise and batch effects in the raw data, cells are not separated well, particularly for dataset 3, i.e., phs001790 (mouse)—GSE115746 (human), and dataset 4, i.e., GSE115746 (mouse)—phs001790 (human). Similarly, in the UMAP projections of Seurat v3 and Conos, cells are not distinguished explicitly in datasets 3 and 4. scmap shows far less discernable results in these paired datasets (Supplementary Fig. 1c). In contrast, when using the aggregated data generated by scGCN, cell subpopulations are clearly discerned in the UMAP projections for all four scenarios.

The UMAP projections suggest that the aggregated cells are aligned better by scGCN than the other methods. This observation is further confirmed by the accuracy score (Fig. 4b). Specifically, scGCN shows the best performance with higher average accuracy (87%), compared with Conos (71%), Seurat v3 (67%), scmap (48%), and CHETAH (20%). We find that Conos shows higher accuracy than Seurat v3 in datasets 1 and 2, but lower in datasets 3 and 4. Quantitatively, in the four paired datasets, scmap and CHETAH produce less accurate results than those of Seurat v3 and Conos. Together, these results show that scGCN performs consistently better in transferring labels (e.g., Seurat v3 and Conos) across different species.

**Transfer labels across datasets of different types of omics.** We next examine how well different algorithms transfer labels across different types of omics. Here we apply scGCN, Seurat v3, and Conos, but omit scmap and CHETAH as they are only designed for scRNA-seq data. Four open accessible paired datasets (A549, brain, kidney, and lung tissues) with scRNA-seq data as the reference and scATAC-seq data as the query dataset, respectively, are included for comparison. We use two evaluation metrics including the batch mixing entropy and the silhouette coefficient.

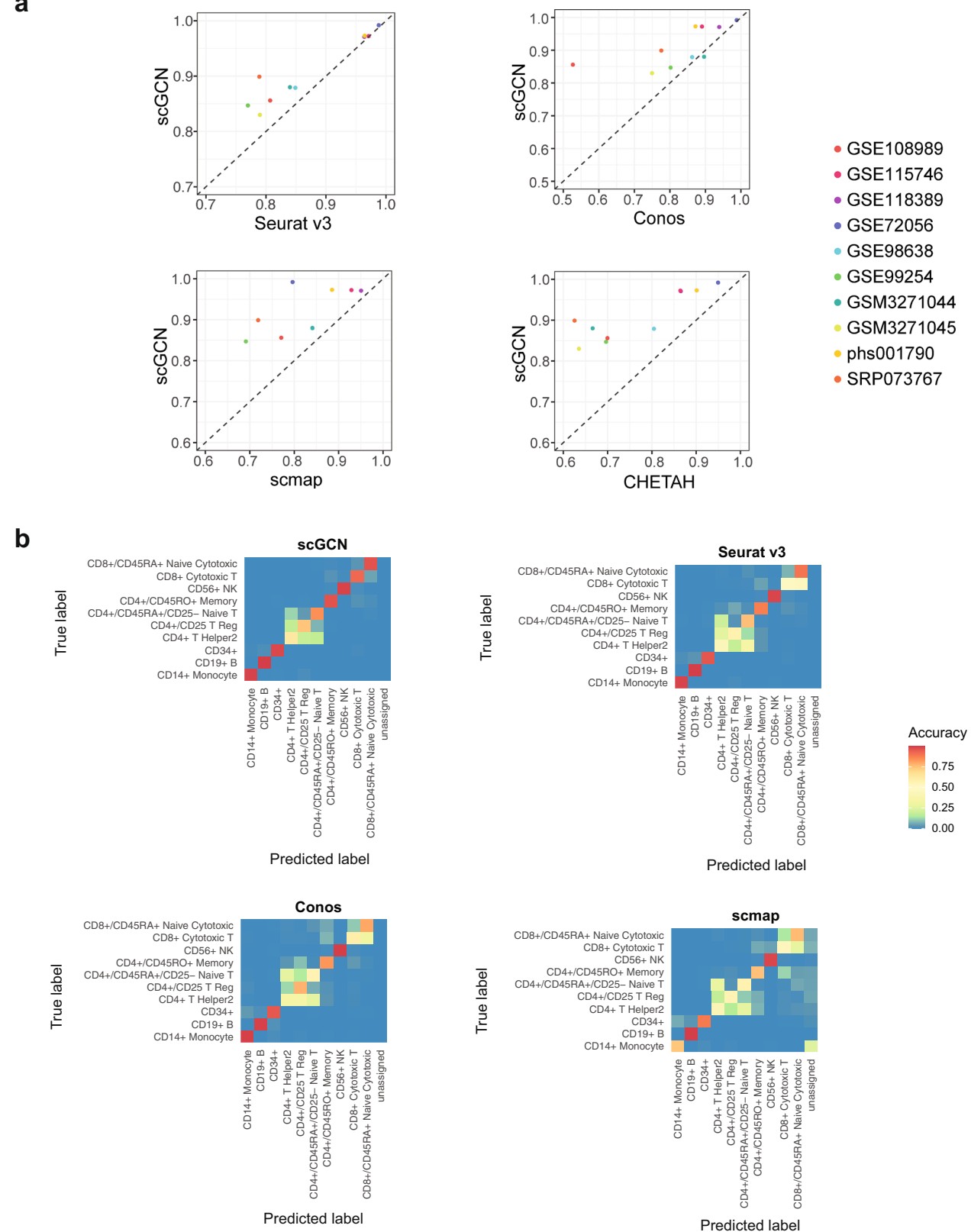

**Fig. 2 Performance of label transfer within datasets. a** Performance of scGCN and other methods (Seurat v3, Conos, scmap, and CHETAH) are measured by the accuracy score on ten datasets. Each point represents the accuracy scores of scGCN versus an alternative method on one dataset. The dashed line represents equivalent accuracy between two methods. Dots above the dashed line indicate that scGCN outperforms the corresponding method on these datasets. **b** Heatmap of the accuracy matrix of each cell type, including different T-cell subtypes, B-cells, and monocytes, which are identified by different methods based on the single cell dataset (SRP073767). Source data are provided as a Source Data file.

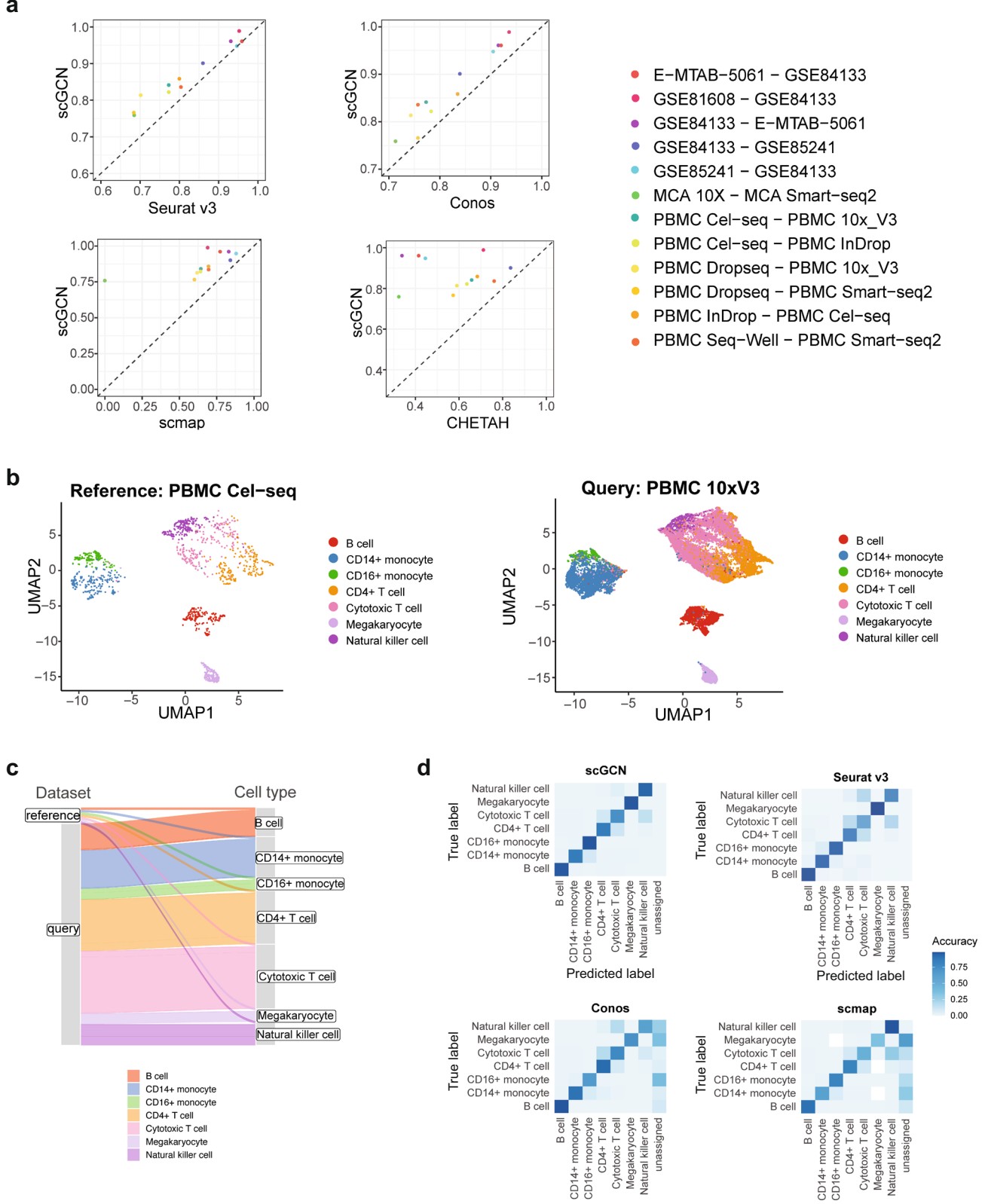

**Fig. 3 Performance of scGCN and its comparison with other methods using 12 cross-platform datasets. a** Performance of scGCN and alternative methods are measured by the accuracy score for 12 paired datasets. Each pair of reference-query datasets is profiled using different single-cell platforms. Each point represents the accuracy score for the query data of each paired datasets. **b** scGCN produces the UMAP projection of cells from Cel-seq as the reference to annotate cells from 10× V2 as the query dataset. **c** The height of linage line in the Sankey diagram reflects the cell number of each cell type in the reference data and query data. **d** Heatmap shows the accuracy matrix of each cell type identified by different methods based on the Cel-seq data as reference and the 10× V2 data as the query dataset. Source data are provided as a Source Data file.

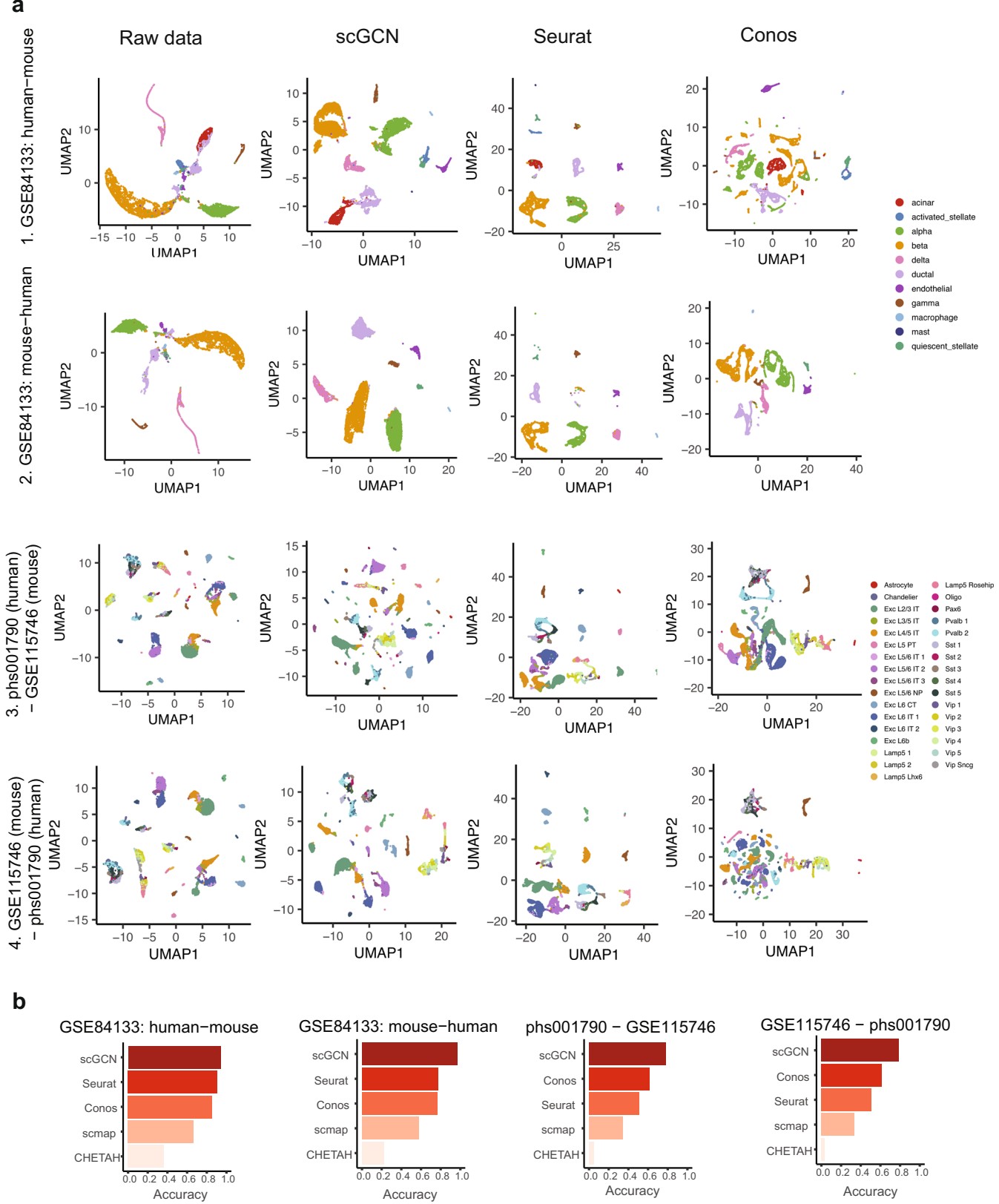

The batch mixing entropy shows the mixing level of cells in the aggregated profiles of reference and query data[38]. A higher entropy value means better intermingling of cells from different batches, whereas the scRNA-seq and scATAC-seq data are regarded as two batches. The silhouette coefficient quantifies the separation of different cell types by calculating the silhouette widths of each cell type[38,39]. A higher silhouette coefficient represents more differences between cell types and fewer variances within each cell type.

Using these two metrics, we evaluate the transferred labels in each query dataset, i.e., scATAC-seq data. Compared to the other methods, scGCN shows better performance with higher mixing

**Fig. 4 Performance of scGCN and its comparison with other methods using four cross-species datasets. a** UMAP projection of four paired cross-species datasets, based on the aggregated data by different methods. The top two rows represent the aggregated data using the human and mouse pancreas dataset GSE84133. First row: human data as the reference and mouse data as the query data. Second row: mouse data as reference and human data as query. Overall, 11 cell type labels are visualized. The bottom two rows represent cells collected from human (phs001790) and mouse cortex (GSE115746), with human and mouse samples as the reference in the third and fourth rows, respectively. Overall, 33 cell type labels are visualized. **b** The bar plots show the performance of scGCN versus other methods (Seurat, Conos, scmap, and CHETAH) that are measured by the accuracy score for four paired cross-species datasets shown in **a**. Each bar represents the accuracy score of each method. Source data are provided as a Source Data file.

entropy (Fig. 5a). Specifically, Seurat v3 has higher mixing entropy than Conos in A549 data, but is lower than Conos in the other three datasets. In the brain dataset, Conos shows comparable mixing entropy with scGCN. We then calculate the silhouette coefficients for all three methods (Fig. 5b). Compared with Seurat v3 and Conos, scGCN has significantly and consistently higher silhouette coefficients. In most datasets, Seurat v3 shows slightly higher silhouette coefficients than Conos.

To further visualize the joint alignment of scRNA-seq data and scATAC-seq data, we show the UMAP projection of the aggregated A549 cells with labels learned by different methods (Fig. 5c). The cell subpopulations (0, 1, and 3 h cells after 100 nM dexamethasone (DEX) treatment) are clearly discerned in the UMAP projection when using the aggregated data generated by scGCN. In contrast, Seurat v3 and Conos can not explicitly distinguish different cell subpopulations (Fig. 5c and Supplementary Fig. 2a).

To characterize the differential accessible loci and uncover the transcriptional regulatory mechanisms in the scATAC-seq data of A549 cells, we perform the motif enrichment analysis to discover the *cis*-regulatory DNA sequences that differentially regulate the 3 h cells after DEX treatment. After treatment, the specific loci of 3 h cells are enriched with the binding motifs of FOXO3, REBB1, and ELF1 compared to 0 h cells (Fig. 5d). Moreover, the transcription factor FOXO3 is upregulated in 3 h cells (Fig. 5e), which is a validated regulator that drives cell progression under the DEX treatment[40,41]. KRT7 and WDR60 also show higher expression in 3 h cells, which occurs in vivo with DEX treatment[42,43] (Supplementary Fig. 2b). RREB1 and ELF1 exhibit upregulation in 3 h cells compared to 0 h cells (Fig. 5e). These results suggest the potential roles of these identified transcriptional factors in maintaining and establishing the chromatin accessibility to express functional genes after treatment. We perform similar motif analysis in mouse brain dataset, based on the integrated scRNA-seq and scATAC-seq datasets from the 10× Genomics Chromium system (Supplementary Fig. 3). In these data, overrepresented DNA motifs are identified in L4-specific accessibility peaks, with Foxp1, Egr3, and Smad3 motifs as the highly enriched motifs (Supplementary Fig. 3a), which also exhibit upregulated expression in L4 cell subtype (Supplementary Fig. 3b). Altogether, these results suggest that scGCN outperforms other methods when transferring labels between single cell transcriptomics and epigenomics data.

## Discussion

Single-cell omics technologies have allowed biologists to gain insights into the individual cellular components of complex biological ecosystems[44–46]. Given the explosive growth in single-cell data, there is a critical need to leverage the existing, well-characterized datasets as references to ensure reliable and consistent annotations of data. In this study, we report a graph-based artificial intelligence model, single-cell Graph Convolutional Network (scGCN) that allows researchers to use reference single-cell omics data to annotate data through robust knowledge transfer approach. We provide evidence that scGCN allows for reliable and reproducible cell type transferring across datasets

from different tissues, species, sequencing platforms, and molecular layers (such as RNA-seq and ATAC-seq). The scGCN software, which is publicly accessible, is compatible with various single-cell datasets for accurate label transfer.

Apart from the technical aspects of computational analysis, the accuracy, robustness, and sensitivity of label transfer also rely on the quality of the reference datasets. Reference datasets used in this study are well-characterized in literature, covering various types of samples and application scenarios in biomedical research. Whereas the current study does not address the quality issues of the reference datasets, it is well accepted that accumulating data in the field will lead to development and definition of higher quality and experimental evidence-based reference datasets. Additionally, when annotating the PBMC 10X V3 data, scGCN reveals accurate cell mapping even when the reference dataset has few cells, suggesting that scGCN is less insensitive to cell number than other methods, which can be further validated with more datasets for a systematic and stringent analysis.

scGCN consistently outperform four commonly used algorithms, namely Seurat v3, Conos, scmap, and CHETAH that possess the knowledge transfer functionalities. scGCN can also overcome batch effects compared to CCA (Supplementary Figs. 4–5 and Supplementary Note 1). In addition to the datasets used in Figs. 2–5, we further include a well-recognized benchmarking collection of single-cell datasets covering 13 major platforms[47] and completely examine all possible reference-query combinations, which allows unbiased evaluation of scGCN (Supplementary Fig. 6–7 and Supplementary Note 2).

With the development of single-cell sequencing technology and growing single-cell data sizes, we show the scalability of scGCN in large-size datasets. Specifically, the memory usage and the computing time are profiled with respect to different sample sizes (from 100 k up to 1 million cells), using the dataset generated by Cao et al[48]. We compare the computational costs of scGCN with Seurat v3 on a computer with 64 GB memory and 3.6 GHz Intel Core i9 processor. We show the computational time and average memory usage for different sample sizes (i.e., 100 k, 200 k, …, 1 millon) in Supplementary Figs. 8–9 and Supplementary Note 3. scGCN is comparable to Seurat v3 in computational time and memory usage[49], demonstrating its efficiency and scalability for large-size single-cell datasets.

From a technical perspective, scGCN has some major advantages. First, unlike the other methods, scGCN simultaneously utilizes features, graphic structures, and reference labels to overcome batch effects, protocol differences, and other intrinsic differences (e.g., different species and omics types) among datasets. Second, scGCN transfers labels from labeled samples to unlabeled samples in a semi-supervised manner, which is more desirable than other unsupervised methods. Third, in every layer, scGCN nonlinearly propagates feature information from neighboring cells in the hybrid graph, which learns the topological cell relations and improves the performance of transferring labels by considering higher-order relations between cells.

Despite these successful results, there are several aspects in which scGCN can be improved. First, as an artificial intelligence (AI) model, scGCN shows not only the merits of its kind, but also

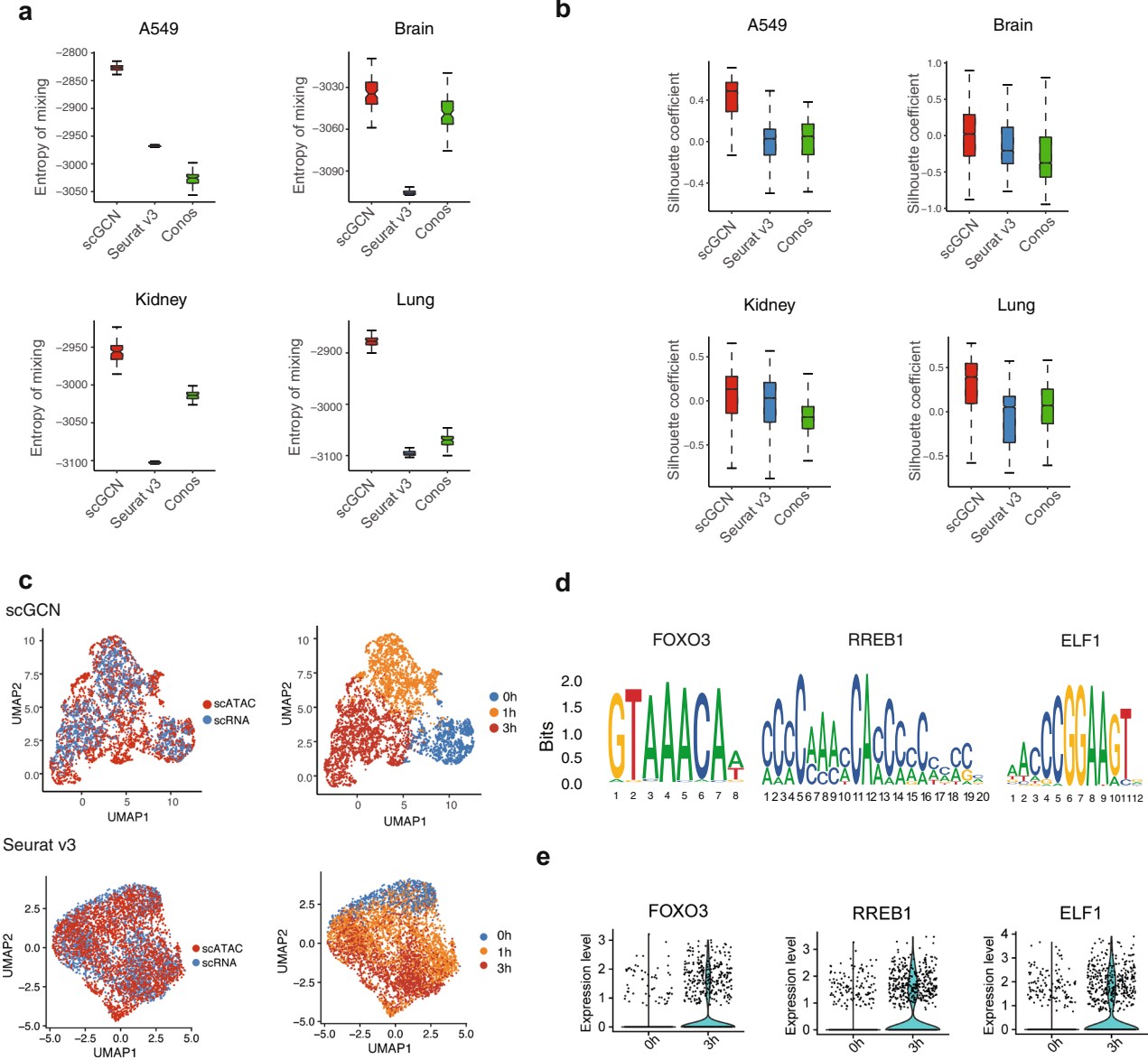

**Fig. 5 Performance of scGCN and its comparison with other methods using four paired cross-omics datasets. a, b** For each method, performance metrics including two indexes, i.e., batch effects entropy (A549 $n = 100$; Brain $n = 100$; Kidney $n = 100$; Lung $n = 100$) and silhouette index (A549 $n = 5282$; Brain $n = 6737$; Kidney $n = 17,674$; Lung $n = 4994$), are used to evaluate the level of cell mixing across datasets and the preservation of local structure within dataset. Data are represented as boxplots where the middle line is the median, the lower and upper hinges correspond to the first and third quartiles, the upper whisker extends from the hinge to the largest value no further than 1.5× the inter-quartile range (IQR) from the hinge, and the lower whisker extends from the hinge to the smallest value at most 1.5× IQR of the hinge. **c** UMAP plots of A549 cells colored by dataset (scRNAseq, scATAC-seq) and by cell states (0, 1, and 3 h), after integration by scGCN and Seurat v3. **d** Overrepresented DNA motifs are identified in 3 h-specific accessibility peaks, with FOXO3, RREB1, and ELF1 motifs as the most highly enriched motifs. **e** These motifs also exhibit upregulated expression in scRNA-seq cells at 3 h. Source data are provided as a Source Data file.

some limitations including the black-box nature of AI models[50–52]. These can be addressed through downstream analysis such as differential gene identification and enrichment analysis, that can ameliorate some of the problems and bring insights into the labeled cells. Second, as a graph model, improving the graph construction can further boost model performance. Our graph construction based on mutual nearest neighbors reflects the state-of-art in single cell graph representation. As a fast-growing research field, graph construction approaches are emerging that we will test and adapt in future versions of scGCN.

## Methods

**Data preprocessing.** For each input data, we denote the dataset with known cell labels as reference data $X_R \in R^{m_0 \times n_r}$ and the dataset that needs to be annotated as query data $X_Q \in R^{m_0 \times n_q}$, where $m_0$ is the number of common gene features shared by $X_R$ and $X_Q$, $n_r$, and $n_q$ are the number of cells in reference data $X_R$ and query data $X_Q$. First, we identify the gene features that exhibit the most variability across different cell types in the reference data, which can be represented as heterogenous and prioritized features. As there are multiple cell types present in the reference data (the number of different cell types is annotated by $F$), we perform multi-class differential expression analysis to $X_R^{m_0 \times n_r}$ using analysis of variance (ANOVA) to identify the most variable genes across different cell types. Bonferroni correction is used to select the top $m = 2000$ genes in reference data with significantly adjusted $P$ values. In scGCN, all input data filters out non-variable gene

features, thus the reference data and the query data become $X_R \in R^{m \times n_r}$ and $X_Q \in R^{m \times n_q}$.

**Graph construction**. We use the mutual nearest neighbor (MNN)[38] approach and canonical correlation analysis[53] to construct a hybrid graph capturing the topological characteristics of single cells in both reference and query datasets. First, the reference dataset as well as the query dataset are standardized. Then a reference-to-query graph (inter-dataset) as well as an internal query graph (intra-dataset) are constructed. The final hybrid graph is composed of these two graphs. The hybrid graph is then used as input for the scGCN model.

**Standardization transformation**. We first perform the standardized transformation for reference data, i.e.,

$$\widetilde{x}_{ij} = \frac{x_{ij} - \bar{x}_i}{\rho_i}, \tag{1}$$

where $x_{ij} \in X_R$ is the raw value of input reference data, $\bar{x}_i$ is the mean of $x_{ij}$, and $\rho_i$ is the standard deviation of $x_{ij}$. Thus $\widetilde{x}_{ij}$ is the standardized value of feature $i$ and cell $j$, where $i \in \{1, 2, \cdots, m\}$ and $j \in \{1, 2, \cdots, n_r\}$. After standardization, reference data $X_R$ is represented as $\widetilde{X}_R$. In the same way, raw data in $X_Q$ is processed in the same manner that becomes $\widetilde{X}_Q$.

**Construction of the reference-to-query graph**. For the scGCN method, one critical step is to construct an effective graph, which is represented as an adjacent matrix that best leverages the reference data and query data. Here, we use the MNN[38] concept and search for the MNNs from both reference and query data after simultaneous dimensionality reduction of reference and query datasets through canonical correlation analysis.

The goal of canonical correlation analysis is to simultaneously project the high-dimensional reference and query data into the same low dimensional space through two dataset-specific linear transformations. Thus, molecular patterns in both reference and query datasets that share the same biological meaning can be captured and represented uniformly in the low dimensional space. $\widetilde{X}_R^{m \times n_r}$ represents the standardized reference data and $\widetilde{X}_Q^{m \times n_q}$ represents the query data. In order to project these two matrixes to a $k$-dimensional space where $k \leq m$, we need to identify $k$ pair of canonical correlation vectors $\mu_i$ of $n_r$ dimension and $\nu_i$ of $n_q$ dimension, where $i = 1, \cdots, k$, to maximize

$$\max_{\mu_i, \nu_i} \mu_i^{\mathrm{T}} \left( \widetilde{X}_R^{m \times n_r} \right)^{\mathrm{T}} \widetilde{X}_Q^{m \times n_q} \nu_i, \tag{2}$$

s.t. $||\mu_i||_2^2 \leq 1$ and $||\nu_i||_2^2 \leq 1$. We use singular value decomposition to calculate the $k$ canonical correlation vector pairs associated with the $k$ largest eigenvalues. Each pair of canonical correlation vector $\mu_i$ and $\nu_i$ thus can be used to project the original data $\widetilde{X}_R^{m \times n_r}$ and $\widetilde{X}_Q^{m \times n_q}$ to the $i$-th dimension in the $k$-dimensional space, respectively. In this study we set $k = 20$.

The reference-to-query graph $A^{RQ}$ is constructed with the MNN approach based on the projected reference and query data. For a cell $i$ in the reference data and another cell $j$ in the query data, $A_{ij}^{RQ} = 1$ if and only if cell $i$ belongs to the nearest neighbors of cell $j$, and at the same time cell $j$ is also one of the nearest neighbors of cell $i$. Otherwise $A_{ij}^{RQ} = 0$. Thus, we construct the inter-dataset cell mapping between reference data and query data.

**Construction of the query internal graph**. Similarly, we also construct internal graph $A^{QQ}$ for cells in the query dataset using the MNN approach and canonical correlation analysis with $k = 20$.

**Construction of the hybrid graph**. The final hybrid graph $A^H \in R^{(n_r+n_q) \times (n_r+n_q)}$ is then constructed by combining the reference-to-query graph ($A^{RQ}$) and the query internal graph ($A^{QQ}$). The hybrid graph $A^H$ is used as the input adjacent matrix for our scGCN model. With this constructed hybrid graph, we validate its effectiveness by comparing it with other different graph construction methods (see Supplementary Figs. 10–12 and Supplementary Note 4).

**scGCN method**. We utilize the GCN[31] for semi-supervised learning and transferring labels from reference data to query data. Each cell is viewed as a node. The annotations of cell types in the reference data are the known labels. The goal of GCN is to predict the cell type annotations of the query data by using not only the features of each cell but also the information leveraging reference and query data, which are characterized as the above hybrid graph $A^H$. Explicitly, the GCN model takes two inputs. One input is the hybrid graph structure learned above, which is represented as the adjacent matrix $A^H \in R^{N \times N}$ (see the "Graph construction" section). The other is the feature matrix $X \in R^{m \times N}$, where $N = n_r + n_q$ is the total number of cells and $m$ is the number of variable features selected in preprocessing. If the reference data is denoted as $X_R \in R^{m \times n_r}$ and the query data as $X_Q \in R^{m \times n_q}$,

the input data matrix is extended to:

$$X = \begin{bmatrix} X_R \\ X_Q \end{bmatrix} \in R^{m \times (n_r+n_q)} \tag{3}$$

Herein, both features of reference and query data, as well as the information leveraging reference and query data are utilized in our model. With these two inputs, the GCN model is constructed with multiple convolutional layers.

For efficient training of GCN introduced in[31], the adjacent matrix $A^H$ is modified as:

$$\widetilde{A} = \check{D}^{-1/2} \hat{A} \check{D}^{-1/2} = \check{D}^{-1/2} (A^H + I) \check{D}^{-1/2}, \tag{4}$$

where $I$ is the identity matrix, $\hat{A} = A^H + I$, and $\check{D}$ is the diagonal degree matrix of $\hat{A}$.

Specifically, each layer is defined as:

$$H^{(l+1)} = f\left(H^{(l)}, \widetilde{A}\right) = \sigma\left(\widetilde{A} H^{(l)} W^{(l)}\right), \tag{5}$$

where $H^{(l)}$ is the input and $W^{(l)}$ is the weight matrix of the $l$-th layer, $\sigma(\bullet)$ is the non-linear activation function, and the input layer $H^{(0)} = X$. The labels of cells in the reference data are represented as a class indicator $y_{lf}$ for cell $l \in \mathscr{Y}_L$ and label $f \in \{1, \cdots, F\}$, where $\mathscr{Y}_L$ represents nodes with known labels, $F$ represents the total number of different labels, and $y_{lf} = 1$ indicates cell $l$ has the $f$-th label, and 0 as not of this label.

Specifically, for a three-layer GCN with $F$ distinct labels, the forward propagation is realized as:

$$\hat{Y} = f(X, A^H) = \mathrm{softmax}\left(\widetilde{A}\,\mathrm{ReLU}\left(\widetilde{A} X^{\mathrm{T}} W^{(0)}\right) W^{(1)}\right), \tag{6}$$

where $W^{(0)} \in R^{m \times h}$ is the input-to-hidden weight matrix projecting the $m$ features input data into an $h$ dimension hidden layer, ReLU stands for the rectified linear unit activation function, $W^{(1)} \in R^{h \times F}$ is a hidden-to-output weight matrix, and $\hat{Y} \in R^{N \times F}$ is the predicted probabilities of cell labels. The softmax activation function is defined as

$$\mathrm{softmax}(\cdot) = \frac{\exp(\cdot)}{\sum \exp(\cdot)}. \tag{7}$$

Given the reference data, we use the cross-entropy error the evaluate the predictions, i.e.,

$$\mathscr{L} = -\sum_{l \in \mathscr{Y}_L} \sum_{f=1}^{F} Y_{lf} \ln\left(\hat{Y}_{lf}\right). \tag{8}$$

After training the model we have

$$\hat{Y} = \mathrm{GCN}(X, A^H) \in R^{(n_r+n_q) \times F}. \tag{9}$$

The prediction $\hat{Y}$ is the probability of each cell within each class. The final label for cell $i$ is determined as the $f$-th label when $\hat{y}_{i,f} \geq 0.5$ where $\hat{y}_{i,f} \in \hat{Y}$.

When applying the scGCN model, we randomly split the reference data as training (80%), test (10%), and validation set (10%), while the query data is unlabeled that can be predicted through the above semi-supervised GCN model. For datasets in this study, we train three-layer scGCN models for a maximum of 200 epochs using Adam with a learning rate of 0.01 and early stopping with a window size of 10. For the number of hidden units, we respectively check 32, 64, 128, 256, 528, 1024, and select the optimal one.

In addition to the semi-supervised GCN[31], we also evaluate the other different graph neural network methods, including HYPERGCN[54], GAT[55], GWNN[56], GraphSAGE[57], and ChebyNet[58], based on our constructed hybrid graph. GCN shows the best overall performance; HYPERGCN, GAT, and GWNN are also good alternatives. We also include the HYPERGCN, GAT, and GWNN models in our scGCN tool to provide more options to users. Results of the comparisons can be found in Supplementary Fig. 13 and Supplementary Note 5.

**Detection of unknown cells**. To identify potential unknown cells in query data, we provide a screening step in our scGCN model using two statistical metrics, entropy score and enrichment score, representing mixtureness and enrichment. Specifically, all cells in query data are grouped to different clusters by modularity-based community detection[59]. For each query cluster, we measure its mixtureness and enrichment based on the inter-data graph of scGCN. Our rationale is that a query cluster of unknown cell type is more likely to have random links to different cell types in the reference data, while a query cluster of known cell type is more likely to link to a specific cell type in the reference data. In this way, unknown cells can be identified by the two statistical metrics. (1) Entropy score: For a cluster $h$ in the query data, the mixtureness of this cluster is defined by the information entropy of normalized enrichment scores. That is,

$$H_h = -\sum_c^{C} \frac{S_{c,h}}{\sum_c^C S_{c,h}} \log \frac{S_{c,h}}{\sum_c^C S_{c,h}} \quad \text{and} \quad S_{c,h} = \frac{m_{c,h}/\sum_c^C m_{c,h}}{n_c/\sum_c^C n_c} \tag{10}$$

where $c$ is a specific cell type, $C$ is the set of all cell types in reference data, $m_{c,h}$ is the number of cells in query cluster $h$ that are linked to cell type $c$ in reference data by the inter-data graph of scGCN, and $n_c$ is the number of cells belonging to cell

type $c$ in reference data. (2) Enrichment score: For a cluster $h$ in the query data, the enrichment score $ES_h$ is defined by the normalized enrichment of the most enriched cell type. That is,

$$ES_h = \max_{c \in C} S_{c,h} / \sum_c^C S_{c,h}. \quad (11)$$

For a cluster in the query data, the entropy score describes whether this cluster in query data is dominated by a specific cell type, and the enrichment score describes how strong this cell type is enriched. Thus, if the query cell cluster $h$ has higher entropy and lower enrichment, these cells should be assigned as unknown cell types. Detailed performance evaluation is shown in Supplementary Fig. 14 and Supplementary Note 6.

**Cross-species classification**. We use the HomoloGene databases provided by NCBI (Build 68) to identify homologous genes between humans and mice, and keep only genes that have a one-to-one correspondence, which serves as a look-up table. After obtaining the intersection gene set between the reference data and the look-up table, the gene names are then converted to human gene names to obtain a compatible input for classification.

**Methods comparison**. For the comparisons with Seurat v3, we used the most recent CRAN release version 3.2.1 of Seurat[26], with its anchoring framework in Seurat v3 for transfer labels across datasets. Specifically, we follow the Seurat v3 vignette about Multiple Dataset Integration and Label Transfer at https://satijalab.org/seurat/v3.2/integration.html, and use the FindTransferAnchors and TransferData functions to transfer cell type labels from a reference dataset onto a query dataset. For the FindTransferAnchors function, we use the default parameters, i.e., reduction = pcaproject, npcs = 30, dims = 1:30, k.anchor = 5, k.filter = 200, k.score = 30, max.features = 200, nn.method = rann, normalization.method = LogNormalize, approx.pca = TRUE. For the TransferData function, we also use the default parameters, i.e., weight.reduction = pcaproject, dims = 1:30, k.weight = 50, sd.weight = 1.

As above, Seurat v3 provides the options of using PCA as the default (PCA-based Seurat v3) and CCA as an alternative (CCA-based Seurat v3) to project the structure of a reference onto the query, which are used in the anchor weighting and label transfer steps[26]. We compare scGCN with CCA-based Seurat v3 in Supplementary Fig. 15 and Supplementary Note 7. For benchmarking with scmap, we identify the classification of query data using the scmapCluster function of scmap. For CHETAH, we identify the classification of query data using the CHETAHclassifier function. All methods are applied with default parameters. In this study, statistical significance is defined using the two-sided Wilcoxon rank test.

**A549 data**. The A549 dataset is downloaded from GEO, which includes 3260 cells profiled using the sci-CAR protocol[60]. scRNA-seq and scATAC-seq data can be accessed with accession number of GSM3271040 and GSM3271041. After quality control, the resultant data consists of 2641 cells in our analysis.

**Mouse brain data**. The single-cell ATAC-seq dataset of adult mouse brain cells is provided by 10× Genomics, which is available through the 10× Genomics website (http://cf.10xgenomics.com/samples/cell-atac/1.1.0/atac_v1_adult_brain_fresh_5k/atac_v1_adult_brain_fresh_5k_filtered_peak_bc_matrix.h5). scRNA-seq data is obtained from the same biological system (the adult mouse brain), which can be downloaded from Allen Institute website (http://celltypes.brain-map.org/api/v2/well_known_file_download/694413985).

**Kidney data**. The kidney dataset is downloaded from GEO, which includes 11,296 single cells from the mammalian kidney using the sci-CAR protocol[60]. The scATAC-seq data and scRNA-seq data that can be accessed with accession number of GSM3271045 and GSM3271044 are profiled using the sci-CAR protocol[60]. Duplicate genes are merged by maximum value and cells that are labeled as NA are removed. Moreover, we remove cells with less than 500 expressed genes and genes expressed in fewer than ten cells from the scRNA-seq data. Cells with less than 200 accessible loci and loci opened in fewer than ten cells are removed from scATAC-seq data. The final dataset with 8837 cells is used in our analysis.

**Lung data**. The lung dataset contains lung count matrices from three scRNA datasets (two lung samples from Tabula Muris [https://figshare.com/projects/Tabula_Muris_Transcriptomic_characterization_of_20_organs_and_tissues_-from_Mus_musculus_at_single_cell_resolution/27733], one lung sample from GSE108097, and two lung sci-ATAC-seq replicates from GSE68103.

**Reporting summary**. Further information on research design is available in the Nature Research Reporting Summary linked to this article.

## Data availability

Source data are provided with this paper. All datasets analyzed in the current study are publicly available and can be downloaded from their public accessions, including GSE108989, GSE115746, GSE118389, GSE72056, GSE98638, GSE99254, GSM3271044, GSM3271045, phs001790, SRP073767 [http://support.10xgenomics.com/single-cell/datasets], E-MTAB-5061, GSE84133, GSE81608, and GSE85241. The PBMC data of six different sequencing protocols are available from the Broad Institute Single Cell portal (https://portals.broadinstitute.org/single_cell/study/SCP424/single-cell-comparison-pbmc-data) and the Zenodo repository (https://doi.org/10.5281/zenodo.3357167)[30]. We select these single-cell data as they have been frequently used to evaluate the performance of transferring labels by different methods[25,26,28,30]. The following shows the details of scRNA-seq and scATAC-seq data used in this study. Source data are provided with this paper.

## Code availability

All the functions mentioned above are implemented as a python software that is available at Github and Zenodo (https://github.com/QSong-github/scGCN)[61].

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

## Acknowledgements

The work is supported in part by the Bioinformatics Shared Resources under the NCI Cancer Center Support Grant to the Comprehensive Cancer Center of Wake Forest University Health Sciences (P30CA012197). WZ is supported by the Hanes and Willis Family Professorship in Cancer. Additional support for Q.S. and W.Z. is provided by a Fellowship to W.Z. from the National Foundation for Cancer Research. This work is also partially supported by the Indiana University Precision Health Initiative to J.S.

## Author contributions

Q.S., J.S., and W.Z. developed the structure and arguments and wrote the manuscript. All the authors reviewed and approved the final manuscript.

## Competing interests

W.Z. is a consultant for Astellas Pharma US, Inc. The other authors have no competing interests to declare.
