## [Peer Review File · Nature Communications]

REVIEWER COMMENTS

Reviewer #1 (Remarks to the Author):

This paper proposes the use of graph convolutional networks to propagate information between samples of single cell data. More specifically, the work here considers a premise where one sample is well annotated, and thus considered as a reference one, while another unlabeled one is considered as a query. Then, the authors construct a graph that combines intra- and inter-sample relations between cells, which form the nodes of the graph. The quantified relation strengths form weighted edges on the graph. Finally, a graph neural networks, using the GCN architecture proposed by Kipf and Welling, is then trained as a semi-supervised node classifier, using the reference annotations as training labels, and its predictions are applied to annotate the remaining nodes in the query-sample portion of the graph.

The proposed approach here makes sense, albeit rather straight forward. Technically, there is limited computational novelty here with respect to the methods used. GCN is applied as-is and highly depends on the graph construction, since single cell data do not generally provide an a priori graph structure. The authors use pretty standard methods to this end, relying on mutual nearest neighbors to determine edges, in conjunction with CCA for aligning different samples to overcome batch effects. It should be noted that numerous other approaches have been proposed in recent years for such alignment. Similarly, various kernel methods have been recently applied to process single cell data (for example, tSNE, UMAP, and PHATE are common visualization methods), and each of these proposes essentially proposes its own graph construction. It seems some ablation study would be appropriate here to verify which construction is suitable for GCN application, and at the same time to establish the robustness of the GCN postprocessing to such constructions.

As far as I know, the use of GCN to propagate labels in this context is new, even though it is well established in other applications, and presented results seem convincing. Therefore, this work could contribute to the readership of Nature Communications who may not be familiar with this approach, although (as mentioned before) the main novelty here is simply in the application of rather standard tools in a new context, rather than the development of a new method. However, if the authors intend to promote the use of graph neural networks, I would suggest they also try to evaluate several other more recent GNNs (such as GraphSAGE or GAT, but there are many others) that have been shown to outperform GCN in a multitude of node classification tasks.

One minor comment I have for the comparisons is concerning the ambiguity in referring to Seurat as a method, while in fact this is a toolbox that contains many different methods for various tasks in it. I guess, from context, the authors used whichever methods are set as the default for given tasks considered here, but it would be more appropriate to refer directly (and cite) the algorithms used rather than the general programming toolbox.

In conclusion, I think this work can be accepted for publication, but I would recommend the authors apply revisions and add ablation studies as described in my comments.

Reviewer #2 (Remarks to the Author):

This manuscript has proposed a self-supervised GCN architecture to label cells according to known

samples. The method have been tested across tissues, species, platforms, and omics types, and shown better accuracy over other methods. Generally, the writing is clear. However, a few issues are required to address.

1. In scGCN model, one critical step is to construct an effective graph where cell nodes are from both reference and query dataset. If there are significant batch effects between the reference and query dataset, constructing the graph based on the raw gene expression with batch effects may be problematic. In the graph construction step of scGCN, canonical correlation analysis (CCA) is used to obtain batch-corrected low-dimensional space where inter-dataset mutual nearest neighbors are identified. Since CCA is a commonly used batch correction method adopted by Seurat and Conos, it is unclear whether the ability of scGCN to overcome batch effects is from CCA or from the semi-supervised GCN model. The authors are suggested to provide more comparisons about the contribution of scGCN for overcoming batch effects.

2. Additionally, the tutorials of Seurat provide the option to use CCA (default is PCA) for integrating reference and query when the batch effects between them are significant. However, in the Methods comparison section, the hyper-parameters of Seurat are not provided. The authors are suggested to provide more details and comparisons since the CCA-based Seurat may have better classification performance than PCA-based Seurat under the setting of large batch difference.

3. The authors didn't introduce how to deal with unknown cell types in query datasets. The measures have ignored new type cells not appearing in the reference cells, will they be assigned to known labels or identified as new types? If wrongly assigned, this may cause problems in many situations. It's necessary to expand the experiments about the accuracy of unknown cell types.

4. They authors didn't describe how to select the reference datasets? The performance of methods are known to highly depend on datasets, and thus the comparisons may be misleading if manually picking datasets from the large number of single cell datasets publicly available.

5. With the development of single-cell sequencing technology, it is worth describing computational issues e.g. memory usage and training time for the method, especially in large-scale data sets.

Reviewers' Comments to Author:

REVIEWER COMMENTS

Reviewer #1 (Remarks to the Author):

This paper proposes the use of graph convolutional networks to propagate information between samples of single cell data. More specifically, the work here considers a premise where one sample is well annotated, and thus considered as a reference one, while another unlabeled one is considered as a query. Then, the authors construct a graph that combines intra- and inter-sample relations between cells, which form the nodes of the graph. The quantified relation strengths form weighted edges on the graph. Finally, a graph neural networks, using the GCN architecture proposed by Kipf and Welling, is then trained as a semi-supervised node classifier, using the reference annotations as training labels, and its predictions are applied to annotate the remaining nodes in the query-sample portion of the graph.

1. The proposed approach here makes sense, albeit rather straight forward. Technically, there is limited computational novelty here with respect to the methods used. GCN is applied as-is and highly depends on the graph construction, since single cell data do not generally provide an a priori graph structure. The authors use pretty standard methods to this end, relying on mutual nearest neighbors to determine edges, in conjunction with CCA for aligning different samples to overcome batch effects. It should be noted that numerous other approaches have been proposed in recent years for such alignment. Similarly, various kernel methods have been recently applied to process single cell data (for example, tSNE, UMAP, and PHATE are common visualization methods), and each of these proposes essentially proposes its own graph construction. It seems some ablation study would be appropriate here to verify which construction is suitable for GCN application, and at the same time to establish the robustness of the GCN postprocessing to such constructions.

Response: Thanks for the reviewer's comments. We do agree that the GCN highly depends on the graph construction. To address the questions raised by the reviewer, we compare different graph construction methods in two aspects: 1) Compare the accuracy of graphs constructed by CCA-MNN and other graph methods (Supplementary Fig. 9); 2) Evaluate the label transfer performance of GCN based on different graph construction methods (Supplementary Fig. 10). After these comparisons, we evaluate the robustness of GCN postprocessing (Supplementary Fig. 11).

Regarding the graph construction methods, we choose three alignment methods that have been proposed recently, including Scanorama¹, scmap-cell², and cellHarmony³. Scanorama first reduces dimensionality using randomized SVD, and then identifies mutual approximate nearest neighbors by locality sensitive hashing to construct the cell alignment graph. scmap-cell builds the alignment graph by performing the approximate nearest neighbor search using product quantization. This alignment graph is further pruned by cosine similarity between neighbored cells with the cutoff threshold of 0.5. cellHarmony first applies Louvain clustering on the k-nearest neighbor graph to identify community partitions, then identifies closest cells between two similar communities from reference and query data, thereby constructs the cell alignment graph.

Additionally, we further choose the kernel-based methods including tSNE⁴, UMAP⁵, and PHATE⁶ as the other three graph construction methods. For tSNE, we construct the graph based on the similarity defined by symmetrized probability distribution using Gaussian kernel over cell pairs, in such a way that similar cells are assigned a higher probability while dissimilar cells are assigned a lower probability. For UMAP, we use the fuzzy graph, *i.e.*, the fuzzy simplicial set of the merged incompatible local views of the input data, which is pruned according to edge membership strength. For PHATE, we construct the graph using the potential distance information, which is computed as a divergence between the associated diffusion probability distributions of the two cells to all other cells, whereby the relationship of each cell to both near neighbors and distant points is accounted for in this distance.

The results from our analyses are summarized below.

1). Compare graph accuracy between CCA-MNN and other graph construction methods

To quantitatively evaluate different graph construction methods, we use the same 26 datasets with true labels in our manuscript to evaluate the performance in three scenarios, *i.e.*, within-data, cross-platform, and cross-species, respectively.

First, for the within-data scenario, we use the same 10 scRNA-seq datasets of Fig. 2. For each dataset, 50% of its cells are randomly selected as the reference data and the other 50% of cells as the query data. Based on the reference-query data, the above six graph construction methods are used to learn both intra-data graph and inter-data graph. To evaluate different graph construction methods, we investigate the inter-graph accuracy and the intra-graph accuracy respectively (Supplementary Fig. 9a), which are defined as the proportions of correctly linked edges among all edges for inter-data graph and intra-data graph respectively.

Supplementary Fig. 9 Accuracy of graphs constructed by different methods. **a** The accuracy of intra-data graph and inter-data graph identified by different methods, including CCA-MNN, Scanorama, scmap-cell, cellHarmony, tSNE, UMAP, and PHATE, within 10 scRNA-seq datasets. Each colored symbol represents the inter-graph accuracy (x-axis) and the corresponding intra-graph accuracy (y-axis) of each method within a certain dataset. Different shapes indicate different methods while different colors represent different datasets. The dashed line shows

equivalent inter-graph accuracy and intra-graph accuracy. CCA-MNN (filled dots above the dashed line) outperforms the other methods on each of these datasets.

As shown in Supplementary Fig. 9a, CCA-MNN consistently demonstrates superior performance across the 10 datasets, with both highest intra-graph accuracy (mean Acc = 93.4%) and inter-graph accuracy (mean Acc = 88.6%). Specifically, CCA-MNN is better than the alignment methods including Scanorama (intra: 82.6%, inter: 77.4%), scmap-cell (intra: 76.4%, inter: 72.9%), and cellHarmony (intra: 93.2%, inter: 73.9%). cellHarmony shows comparable accuracy with CCA-MNN on the intra-dataset graph, but not inter-dataset graph. Moreover, CCA-MNN also outperforms the kernel-based methods, *i.e.*, UMAP (intra: 68.2%, inter: 71.2%), tSNE (intra: 73.4%, inter: 75.3%), and PHATE (intra: 70.0%, inter: 73.6%). UMAP and PHATE show particularly lower accuracy relative to the other methods, especially in datasets GSE98638 and GSE99254. Interestingly, the overall accuracy pattern of all methods is aligned along the dashed line, indicating that intra-graph accuracy differs not much from inter-graph accuracy when there are no batch effects between reference and query data that are generated from the same dataset.

Second, we evaluate these graph construction methods using cross-platform datasets. Here we use the same 12 paired reference-query datasets as in Fig. 3. The reference and query data in each pair are profiled by different scRNA-seq technologies. We then apply the above methods to learn the intra-data graph and inter-data graph respectively. Similarly, we use both intra-graph accuracy and inter-graph accuracy to evaluate the performance of each graph construction method.

Supplementary Fig. 9 Accuracy of graphs constructed by different methods. b Performance of CCA-MNN and other graph construction methods (Scanorama, scmap-cell, cellHarmony, tSNE, UMAP, PHATE) are measured by the accuracy of intra-data graph and inter-data graph based on cross-platform datasets. Each colored symbol represents the inter-graph accuracy (x-axis) and the corresponding intra-graph accuracy (y-axis) of each method on a certain reference-query pair. Different shapes indicate different methods while different colors represent different datasets. The dashed line shows equivalent inter-graph accuracy and intra-graph accuracy. CCA-MNN (filled dots above the dashed line) outperforms the other methods on each of these datasets.

As shown in Supplementary Fig. 9b, CCA-MNN consistently demonstrates better performance than other methods across the 12 pairs of datasets, with both higher intra-graph accuracy (mean

Acc = 97.9%) and inter-graph accuracy (mean Acc = 88.6%). Specifically, CCA-MNN is higher than the alignment methods including Scanorama (intra: 90.9%, inter: 76.4%), scmap-cell (intra: 90.2%, inter: 79.2%), and cellHarmony (intra: 96.9 %, inter: 69.5%). cellHarmony shows similar performance as CCA-MNN regarding the intra-dataset graph, but not the inter-dataset graph. Moreover, CCA-MNN also outperforms the kernel-based methods, *i.e.*, UMAP (intra: 83.3%, inter: 76.7%), tSNE (intra: 80.5%, inter: 67.8%), and PHATE (intra: 58.9%, inter: 37.1%). PHATE shows lower accuracy relative to other methods in certain reference-query pairs (*e.g.*, GSE81608 - GSE84133 and PBMC Seq-well - PBMC Smart-Seq2). Interestingly, the alignment methods tend to perform better than the kernel-based methods, except that UMAP shows comparable accuracy with the alignment methods. Different from the within-data scenario (Supplementary Fig. 9a), the overall accuracy pattern of all methods is shifting left from the dashed line, suggesting the substantial batch effects between reference and query data.

Supplementary Fig. 9 Accuracy of graphs constructed by different methods. *c* CCA-MNN and other graph construction methods (Scanorama, scmap-cell, cellHarmony, tSNE, UMAP, PHATE) are evaluated by the intra-graph accuracy and inter-graph accuracy based on cross-species datasets. Each symbol represents the inter-graph accuracy (x-axis) and the corresponding intra-graph accuracy (y-axis) of each method on a certain dataset. Different shapes indicate different methods while different colors represent different datasets. The dashed line shows equivalent inter-graph accuracy and intra-graph accuracy. CCA-MNN (filled dots above the dashed line) outperforms the other methods on each of these datasets.

Lastly, we compare the graph construction methods using datasets from different species. Here we use the same four reference-query pairs of cross-species datasets as in Fig. 4. From Supplementary Fig. 9c, we find that all methods show comparably high intra-graph accuracy when leveraging human and mouse data from GSE84133, *i.e.*, CCA-MNN (intra: 99.5%, inter: 91.9%), Scanorama (intra: 97.9%, inter: 90.7%), scmap-cell (intra: 97.7%, inter: 77.6%), cellHarmony (intra: 98.0%, inter: 90.6%), UMAP (intra: 95.7%, inter: 79.1%), tSNE (intra: 93.3%, inter: 60.9%), and PHATE (intra: 97.1%, inter: 83.6%). However, CCA-MNN presents decent inter-graph accuracy (intra: 97.5%, inter: 72.0%) when identifying graphs between phs001790 and GSE115746 datasets, whereas the other methods show much lower inter-graph accuracy, including Scanorama (intra: 94.9%, inter: 31.2%), scmap-cell (intra: 87.3%, inter: 38.5%), cellHarmony (intra: 98.5%, inter: 44.7%), UMAP (intra: 92.7%, inter: 21.9%), tSNE (intra: 92.8%, inter: 18.9%), and PHATE (intra: 97.2%, inter: 6%). Such lower inter-graph

accuracy may be due to the substantial batch effects between phs001790 and GSE115746 datasets. This can be observed from Supplementary Fig. 9c that data points of phs001790 - GSE115746 pairs are driven far away from the dashed line.

Altogether, the above results (Supplementary Fig. 9) show that CCA-MNN performs consistently better in identifying accurate hybrid graphs within data, cross platforms, and cross species.

2). Evaluate the label transfer performance of GCN based on different graph construction methods

After the comparisons of graph accuracy, we further evaluate the performance of GCN based on different graph construction methods. Here we use the same 26 datasets in our manuscript to evaluate the performance in three scenarios, *i.e.*, within-data, cross-platform, and cross-species, respectively.

Supplementary Fig. 10 Performance of GCN based on different constructed graphs. a Performance of GCN using different graph construction methods, *i.e.*, CCA-MNN+GCN, Scanorama+GCN, scmap-cell+GCN, cellHarmony+GCN, tSNE+GCN, UMAP+GCN, and PHATE+GCN, is measured by the accuracy score within the 10 scRNA-seq datasets. Each shape with corresponding color represents each of the different methods. Solid line is shown above the dashed lines demonstrating that scGCN outperforms the other methods across these datasets.

First, we use the same 10 scRNA-seq datasets whereby 50% of cells in each dataset are randomly selected as the reference data and the other 50% of cells as the query data. We evaluate the performance of GCN based on different graph construction methods by the accuracy score (Acc), which is defined as the proportion of correctly predicted cells among all cells in the query data. As shown in Supplementary Fig. 10a, CCA-MNN+GCN (*i.e.*, scGCN) consistently demonstrates better performance with higher accuracy (mean Acc = 90.9%) than other methods across datasets. Specifically, CCA-MNN+GCN is higher than the alignment-based methods including Scanorama+GCN (86.0%), scmap-cell+GCN (84.3%), and cellHarmony+GCN (81.7%). CCA-MNN+GCN also outperforms the kernel-based methods, *i.e.*, UMAP+GCN (79.1%), tSNE+GCN (acc=80.5%), and PHATE+GCN (77.4%). Consistent with the results of

graph accuracy in Supplementary Fig. 9a, UMAP and PHATE show relatively lower performance on datasets GSE98638 and GSE99254.

Supplementary Fig. 10 Performance of GCN based on different constructed graphs. b Performance of GCN using different graph construction methods (CCA-MNN+GCN, Scanorama+GCN, scmap-cell+GCN, cellHarmony+GCN, tSNE+GCN, UMAP+GCN, PHATE+GCN) is measured by the accuracy score using 12 paired cross-platform datasets. Each colored symbol represents each of the different methods. Solid line is above the dashed line showing that CCA-MNN+GCN outperforms the other methods on most of the paired datasets.

Second, we compare these methods on the reference-query datasets from different experimental platforms. Here we include the same 12 paired reference-query datasets used in Fig. 3. Each pair of reference-query datasets are profiled by different scRNA-seq technologies. Evaluated by the accuracy score, all methods present decent performance (Supplementary Fig. 10b), whereas CCA-MNN+GCN outperforms with consistently higher accuracy (mean Acc = 87.1%) than the alignment-based methods including Scanorama+GCN (82.9%), scmap-cell+GCN (85.9%), and cellHarmony+GCN (83.5%). These alignment-based methods achieve approximate performance with the kernel-based methods including UMAP+GCN (83.8%), tSNE+GCN (82.2%), but not PHATE+GCN (76.2%). These results suggest that intra-graph, inter-graph, and GCN postprocessing all play important roles for accurate prediction.

Supplementary Fig. 10 Performance of GCN based on different constructed graphs. c Performance of GCN using different graph construction methods (CCA-MNN+GCN, Scanorama+GCN, scmap-cell+GCN, cellHarmony+GCN, tSNE+GCN, UMAP+GCN, PHATE+GCN) is measured by the accuracy score using the 4 paired cross-species datasets. Colored symbols represent different methods. Solid line located above the dashed line represents that scGCN outperforms the other methods across these paired datasets.

Lastly, we compare these methods on the reference-query datasets from different species. Here we keep using the same four pairs of cross-species datasets as in Fig. 4. In Supplementary Fig. 10c, all methods show high performance when leveraging human and mouse data from GSE84133 (CCA-MNN+GCN: 95.5%; Scanorama+GCN: 91.6%; scmap-cell+GCN: 92.4%; cellHarmony+GCN: 92.3%; UMAP+GCN: 90.6%; tSNE+GCN: 93.3%; PHATE+GCN: 90.4%), which is consistent with their corresponding graph accuracy (Supplementary Fig. 9c). However, CCA-MNN+GCN presents adequate accuracy for predicting the labels of phs001790 or GSE115746 datasets, whereas the other methods show lower performance, which is also consistent with their inter-graph accuracy (Supplementary Fig. 9c).

3). Evaluate the robustness of GCN postprocessing

Supplementary Fig. 11 Robustness of GCN postprocessing based on CCA-MNN constructed graphs. For each reference-query pair from different scenarios, *i.e.*, within-data, cross-platform, and cross-species, we use 10 random initializations of scGCN model and examine the accuracy scores accordingly.

After comparing with the above graph construction methods, we further establish the robustness of the GCN postprocessing to the graph constructed by CCA-MNN. Here we include the total 26 paired reference-query datasets used as above, *i.e.*, within-data, cross-platform, and cross-species. For each reference-query pair, we first use CCA-MNN to construct hybrid graph and then repeatedly train GCN model using 10 different training/val/test set that are randomly split from reference data. Boxplots in Supplementary Fig. 11 show the stable accuracy of GCN based on the constructed graph in each pair of reference-query datasets. Different colors represent different scenarios of paired datasets. The average standard deviation of accuracy scores is 0.0017, 0.0038, and 0.0049 respectively in the three scenarios. This result confirms the robustness of GCN postprocessing to the CCA-MNN constructed graph.

In summary, through the comparisons of CCA-MNN with other graph construction methods based on both 1) the graph accuracy, 2) the label transfer performance, as well as 3) the robustness of GCN postprocessing based on CCA-MNN, our method shows superior and robust performance when transferring labels in within-data, cross-platform, and cross-species scenarios. The related revisions can be found in the updated manuscript Page 8 and the Supplementary Note 4.

2. As far as I know, the use of GCN to propagate labels in this context is new, even though it is well established in other applications, and presented results seem convincing. Therefore, this

work could contribute to the readership of Nature Communications who may not be familiar with this approach, although (as mentioned before) the main novelty here is simply in the application of rather standard tools in a new context, rather than the development of a new method. However, if the authors intend to promote the use of graph neural networks, I would suggest they also try to evaluate several other more recent GNNs (such as GraphSAGE or GAT, but there are many others) that have been shown to outperform GCN in a multitude of node classification tasks.

Response: We appreciate the reviewer's positive comments and encouragement. Following the reviewer's suggestion, we choose six different graph neural networks for evaluation. They are Graph Attention Network (GAT)⁷ that uses the multi-head self-attention mechanism to learn the hidden representations of graph-structured data for node classification; GraphSAGE⁸ with LSTM and max-pooling aggregators that uses stochastic neighborhood sampling and aggregation for representation learning on large graphs; ChebyNet that convolutes graph-structured data using fast localized spectral filtering⁹; a most recently proposed method HYPERGCN¹⁰ that maps node features from Euclidean to hyperbolic space and implements hyperbolic graph convolution to learn inductive model for node representations; and another recent developed method, GWNN¹¹ that uses graph wavelet transform convolution operator rather than graph Fourier transform on graph data.

According to their published results, the above methods have been compared with GCN on semi-supervised classification of three benchmark datasets: Cora, Citeseer and Pubmed. Specifically, GAT outperforms GCN with 1.6% on Cora and Citeseer but not Pubmed dataset⁷. For GraphSAGE-LSTM and GraphSAGE-maxpool, we do not find their direct comparisons with GCN in literature. ChebyNet shows respectively 0.3%, 0.5%, and 4.6% lower accuracy than GCN on Cora, Citeseer and Pubmed data⁷. HYPERGCN is shown with 2% higher accuracy than GCN on Pubmed data but has 2% lower accuracy than GCN on Cora dataset¹⁰. HYPERGCN does not show its performance on Citeseer dataset. GWNN is shown to outperform GCN with 1.3% on Cora and Citeseer datasets but not Pubmed dataset¹¹. These reports show that, even though these methods use complicated formulation for graph convolution, their performance is marginally improved or even lower than GCN in these benchmarking datasets. Meanwhile, these graph neural network methods have not been evaluated for label transfer across different datasets.

To gain insights into their label transfer performance across different single-cell datasets, we compare these methods quantitatively. Here we use the same 26 datasets with true labels in our manuscript to evaluate their performance. That is, for comparisons in the within-data scenario, we use the same 10 scRNA-seq datasets from Fig. 2. For each dataset, 50% cells are randomly selected as the reference data and the other 50% cells as the query data. For comparisons in the cross-platform scenario, we use the same 12 paired reference-query datasets as in Fig. 3. Each pair of reference-query datasets are profiled using different scRNA-seq technologies. For comparisons in the cross-species scenario, we use the same four pairs of cross-species datasets from Fig. 4. The performance of each method is evaluated by the accuracy score (Acc).

The comparison results are summarized in Supplementary Fig. 12. Each panel shows the accuracy scores of scGCN versus one of the six methods on all 26 pairs of datasets. Different

colors represent different paired reference-query datasets, while different symbols represent different label transfer scenarios, *i.e.*, within-data, cross-platform, cross-species.

Supplementary Fig. 12 Performance of label transfer using different graph neural networks

Performance of scGCN and other methods including HYPERGCN (a), GAT (b), GWNN (c), GraphSAGE-maxpool (d), GraphSAGE-LSTM (e), and ChebyNet (f) is measured by the accuracy score on all 26 datasets. Different colors represent different paired reference-query datasets, while different symbols represent different label transfer scenarios, *i.e.*, within-data, cross-platform, cross-species. The dashed line represents equivalent accuracy between two methods.

Supplementary Fig. 12a shows the accuracy scores of each paired datasets for HYPERGCN versus scGCN respectively. For HYPERGCN, we use the ‘HGCN’ model with ‘PoincareBall’ manifold as it shows the best performance. The overall accuracy of HYPERGCN does not show much difference with scGCN. The average difference of accuracy between HYPERGCN and scGCN is

1.02%. The most different performances between HYPERGCN and scGCN are shown on two datasets, *i.e.*, GSM3271044 and phs001790 (human) – GSE115746 (mouse). For the GSM3271044 dataset, HYPERGCN outperforms scGCN with 3.15%. For the phs001790 (human) – GSE115746 (mouse) dataset, HYPERGCN shows 5.58% less performance than scGCN. When transferring labels across all 26 datasets, HYPERGCN outperforms scGCN in 10 datasets. The mean accuracy of HYPERGCN on all datasets is 88.14%, which is slightly lower than scGCN (88.57%).

Supplementary Fig. 12b shows the accuracy scores of scGCN and GAT respectively based on each paired reference-query dataset. For GAT, we use 32 attention heads instead of the default 8 for better performance, and the same number of hidden units per each attention head in each layer as scGCN. The mean accuracy of GAT on all 26 datasets is 87.22%, which is a bit lower than scGCN (mean Acc = 88.57%). Among the 26 datasets, GAT slightly outperforms scGCN in 7 cases. Specifically, GAT shows better performance than scGCN on 6 of the 12 cross-platform datasets (increased accuracy of GAT over scGCN), *i.e.*, MCA 10x – MCA Smart-Seq2 (1.32%), E-MTAB-5061 – GSE84133 (0.28%), GSE81608 – GSE84133 (0.035%), PBMC DropSeq – PBMC10xV3 (1.09%), PBMC Cel-seq – PBMC InDrop (2.60%), PBMC InDrop – PBMC Cel-Seq (1.47%). Additionally, for the cross-species data, GAT is better than scGCN on datasets (GSE84133: mouse – human) with 0.7% increase of accuracy.

Supplementary Fig. 12c shows the accuracy scores of scGCN and GWNN respectively on each paired dataset. For GWNN, we use the same numbers of hidden units in each layer as scGCN. Interestingly, we find that the accuracy of GWNN is comparable with scGCN on each of the 26 datasets with minimal difference. The mean accuracy of GWNN on all 26 datasets is 87.28%, which is a bit lower than scGCN (mean Acc= 88.57%) but higher than GAT (mean Acc= 87.22%). Specifically, GWNN shows small increase of accuracy than scGCN on 6 of the 12 cross-platform datasets (increased accuracy of GWNN over scGCN), *i.e.*, MCA 10x – MCA Smart-Seq2 (0.767%), GSE85241 – GSE84133 (0.931%), E-MTAB-5061 – GSE84133 (0.103%), PBMC DropSeq – PBMC Smart-Seq2 (0.0869%), PBMCInDrop – PBMC Cel-Seq (0.549%). GWNN is shown underperform scGCN in other scenarios.

The other three methods show obviously lower accuracy than scGCN (Supplementary Fig. 12d-f). Supplementary Fig. 12d and Supplementary Fig. 12e show the accuracy scores of scGCN versus GraphSAGE-maxpool and GraphSAGE-LSTM respectively. We observe that scGCN has higher accuracy than both GraphSAGE-maxpool and GraphSAGE-LSTM models in almost all datasets, except that GraphSAGE-maxpool is 0.086% higher than scGCN in “PBMC DropSeq – PBMC Smart-Seq2”, and GraphSAGE-LSTM is 1.79% higher than scGCN in GSM3271044 dataset. The mean accuracy of GraphSAGE-maxpool and GraphSAGE-LSTM on all 26 datasets is 77.50% and 77.7% respectively. Finally, for ChebyNet, the accuracy scores of scGCN versus ChebyNet are shown in Supplementary Fig. 12f. ChebyNet is applied with same parameters as scGCN. ChebyNet shows less accuracy in all datasets except the GSE72056 dataset, of which ChebyNet outperforms scGCN with 0.0193%. The mean accuracy of ChebyNet on all 26 datasets is 78.75 % that is lower than scGCN.

In the above comparisons, the six graph neural network methods do not show superior performance than scGCN. This may be due to 1) single-cell data with constructed graphs is different from their paper’s benchmarking data (Cora, Citeseer, and Pubmed) where the graph

originally exists; 2) the performance is evaluated for label transfer across two different datasets. Altogether, our comparisons demonstrate that graph neural network methods perform very well for single-cell label transfer, which also outperform current methods (Fig. 2-4). Among all network methods, scGCN shows best overall performance, meanwhile HYPERGCN, GAT, and GWNN are also good alternatives. For user convenience, we also include the HYPERGCN, GAT, and GWNN models in our scGCN tool to provide more options to users. We add the revisions in the updated manuscript Page 10 and the Supplementary Note 5.

3. One minor comment I have for the comparisons is concerning the ambiguity in referring to Seurat as a method, while in fact this is a toolbox that contains many different methods for various tasks in it. I guess, from context, the authors used whichever methods are set as the default for given tasks considered here, but it would be more appropriate to refer directly (and cite) the algorithms used rather than the general programming toolbox.

Response: Thanks for the reviewer's comments. We agree that Seurat is a general toolbox that is ambiguous in referring to a specific method. Since the anchor-based label transfer method is a major new method proposed in Seurat v3¹², it has been broadly referred as Seurat v3 in¹² and other literature^{13,14} in such context. Therefore, we follow this convention, refer this label transfer method as Seurat v3, and revise our manuscript accordingly.

We specify this label transfer method in the introduction (updated manuscript Page 2):

Seurat is a well-established, widely used toolkit for single cell genomics^{12,15}. Recently a new anchor-based label transfer method across substantially different single-cell samples has been proposed in Seurat v3¹².

We also add the following description in the 'Methods comparison' section (updated manuscript Page 11) to further clarify:

For the comparisons with Seurat, we used the most recent CRAN release version 3.2.1 of Seurat¹², with its new anchoring framework in Seurat v3 for transfer labels across datasets. Specifically, we follow the Seurat v3's vignette "Multiple Dataset Integration and Label Transfer" at <https://satijalab.org/seurat/v3.2/integration.html>, and use the "FindTransferAnchors" and "TransferData" functions to transfer cell type labels from a reference dataset onto a new query dataset. For the "FindTransferAnchors" function, we use the default parameters, *i.e.*, reduction = "pcaproject", npcs = 30, dims = 1:30, k.anchor = 5, k.filter = 200, k.score = 30, max.features = 200, nn.method = "rann", normalization.method = "LogNormalize", approx.pca = TRUE. For the "TransferData" function, we also use the default parameters, *i.e.*, weight.reduction = "pcaproject", dims = 1:30, k.weight = 50, sd.weight = 1.

In conclusion, I think this work can be accepted for publication, but I would recommend the authors apply revisions and add ablation studies as described in my comments.

Response: Thank you for your support! We hope you would agree that we have improved our analyses according to your constructive suggestions.

Reviewer #2 (Remarks to the Author):

This manuscript has proposed a self-supervised GCN architecture to label cells according to known samples. The method have been tested across tissues, species, platforms, and omics types, and shown better accuracy over other methods. Generally, the writing is clear. However, a few issues are required to address.

1. In scGCN model, one critical step is to construct an effective graph where cell nodes are from both reference and query dataset. If there are significant batch effects between the reference and query dataset, constructing the graph based on the raw gene expression with batch effects may be problematic. In the graph construction step of scGCN, canonical correlation analysis (CCA) is used to obtain batch-corrected low-dimensional space where inter-dataset mutual nearest neighbors are identified. Since CCA is a commonly used batch correction method adopted by Seurat and Conos, it is unclear whether the ability of scGCN to overcome batch effects is from CCA or from the semi-supervised GCN model. The authors are suggested to provide more comparisons about the contribution of scGCN for overcoming batch effects.

Response: Thanks for the reviewer's comments. To illustrate the contributions of scGCN for overcoming batch effects, we provide comparisons in two aspects. First, we evaluate the capability of CCA and scGCN in reducing batch effects while preserving cell type differences between datasets. Second, we evaluate the ability of CCA and scGCN for overcoming batch effects by their label transfer performance.

1). Reducing batch effects while preserving cell type differences

First, to illustrate the contributions of scGCN for overcoming batch effects, we add comparisons with CCA on datasets of within-data, cross-platform, and cross-species scenarios that have true labels. Here we use both batch mixing entropy (Supplementary Fig. 4a) and cell-type mixing entropy (Supplementary Fig. 4b) as the evaluation indexes. Specifically, the batch mixing entropy shows the mixing level of cells from reference and query data¹⁶. A higher batch mixing entropy indicates better intermingling of cells from different data batches, wherein the reference and query data are regarded as two batches. The cell-type mixing entropy quantifies the intermingling of cells from different cell types¹⁶. A lower cell-type mixing entropy represents better separation of different cell types. Therefore, using these two evaluation indexes, we can measure the contributions of CCA and scGCN for eliminating batch effects while preserving the differences between different cell types.

Supplementary Fig. 4a shows the batch mixing entropy of scGCN and CCA respectively based on each paired reference-query dataset. Specifically, for within-data scenario that has no batch effects, CCA and scGCN both show higher batch mixing entropy. When applying to the reference-query pairs from different platforms and species, CCA and scGCN both show lower mixing entropy, indicating the underlying batch effects. Across all datasets, the batch mixing entropy of CCA does not show significant differences with that of scGCN (P value = 0.374). It suggests that scGCN has comparable capability with CCA in reducing batch effects from different datasets. Supplementary Fig. 4b shows the cell-type mixing entropy of scGCN and CCA respectively. We find that scGCN shows much lower cell-type mixing entropy than CCA, demonstrating that scGCN achieves better cell type separation. Across all datasets, the cell-type mixing entropy of scGCN is significantly lower than CCA (P value = $3.632e-12$). It suggests that

scGCN preserves cell type differences when applying to different datasets. Overall, these results show that CCA only mixes cells from different data batches but cannot retain cell type differences. In contrast, scGCN not only reduces batch effects but also avoids mixing cells from different cell types, demonstrating the contributions of our scGCN method in overcoming batch effects.

Supplementary Fig. 4 Batch effects evaluation of CCA and scGCN. CCA and scGCN are evaluated by the batch mixing entropy (a) and the cell-type mixing entropy (b) respectively. Different colors represent different paired reference-query datasets, while different symbols represent different label transfer scenarios, i.e., within-data, cross-platform, cross-species. The dashed line represents equivalent mixing entropy between two methods.

2). Evaluating ability of overcoming batch effects by label transfer performance

As mentioned by the reviewer, in the graph construction step of scGCN, canonical correlation analysis (CCA) is used to obtain batch-corrected low-dimensional space where inter-dataset mutual nearest neighbors (MNN) are identified. Here we use the CCA-MNN to identify the transferred labels, and compare with scGCN in their abilities of overcoming batch effects.

For CCA-MNN, we identify the transferred labels based on its graph. That is, for each cell in query data, its label is determined by its linked cells in reference data, according to the CCA-MNN graph. In this way, the ability of CCA-MNN in overcoming batch effects for label transfer

can be measured by the accuracy of identified cell labels in the query data. In Supplementary Fig. 5, we show the specific contributions of CCA-MNN and scGCN, using 26 datasets from three scenarios that have true labels, *i.e.*, within-data (Supplementary Fig. 5a), cross-platform (Supplementary Fig. 5b), and cross-species (Supplementary Fig. 5c). For each paired data, blue colored bar indicates the accuracy of CCA-MNN, and yellow bar indicates the increased accuracy of scGCN than CCA-MNN.

Supplementary Fig. 5 Label transfer accuracy of CCA-MNN and scGCN. Three panels **a - c** show the accuracy of labels identified by CCA-MNN and scGCN, based on the reference-query pairs from within-data (**a**), cross-platform (**b**), and cross-species datasets (**c**), respectively. The blue-colored bar represents the accuracy of CCA-MNN, while the yellow-colored bar represents the improved accuracy from the post-processing GCN model.

Supplementary Fig. 5a shows the accuracy of CCA-MNN and scGCN based on the 10 scRNA-seq datasets from within-data scenario (same data used in Fig. 2), whereby 50% of cells in each dataset are randomly selected as the reference data and the other 50% of cells as the query data. We find that across the 10 datasets, scGCN exceeds CCA-MNN with an average of 24.45% in accuracy, showing that the semi-supervised GCN component brings significant increase of accuracy, with the most increases observed in three datasets: SRP073767 with 53.36% increase, GSM3271044 with 44.96% increase, and GSM3271045 with 36.95% increase.

Supplementary Fig. 5b shows the accuracy of CCA-MNN and scGCN using the 12 reference-query pairs from the cross-platform datasets (same datasets with Fig. 3). We find that the semi-supervised GCN component of scGCN brings an average 60.83% increase of accuracy than CCA-MNN, with much more improvement than the within-data scenario (Supplementary Fig. 5a). CCA-MNN shows the best performance on three paired datasets, including GSE84133 – GSE85241 with 74.48% accuracy, GSE84133 – E-MTAB-5061 with 74.15% accuracy, and MCA 10x – MCA Smart-Seq2 with 61.67% accuracy, which are consistent with their less batch effects shown in Supplementary Fig. 4.

Supplementary Fig. 5c shows the accuracy of CCA-MNN and scGCN using the 4 reference-query pairs from different species (same datasets with Fig. 4). The yellow bar shows the increased accuracy contributed by the semi-supervised GCN component in scGCN. Similarly, CCA-MNN doesn't show decent accuracy in aligning datasets with substantial batch effects, with an average of 65.50% less accuracy than scGCN. The best accuracy of CCA-MNN is 50.99% when transferring labels across reference-query pair “GSE84133: human – mouse”.

Overall, from the above comparisons, we find that CCA-MNN is not sufficient to overcome batch effects. The above results demonstrate the contributions of semi-supervised GCN model in overcoming batch effects and label transfer. We add the related revisions in the updated manuscript Page 6 and the Supplementary Note 1.

2. Additionally, the tutorials of Seurat provide the option to use CCA (default is PCA) for integrating reference and query when the batch effects between them are significant. However, in the Methods comparison section, the hyper-parameters of Seurat are not provided. The authors are suggested to provide more details and comparisons since the CCA-based Seurat may have better classification performance than PCA-based Seurat under the setting of large batch difference.

Response: Thanks for the reviewer's comments.

In the 'Methods comparison' section (updated manuscript Page 11), we provide the hyper-parameters of Seurat and revise the manuscript as below:

For the comparisons with Seurat, we used the most recent CRAN release version 3.2.1 of Seurat¹², with its new anchoring framework in Seurat v3 for transfer labels across datasets. Specifically, we follow the Seurat v3's vignette “Multiple Dataset Integration and Label Transfer” at <https://satijalab.org/seurat/v3.2/integration.html>, and use the “FindTransferAnchors” and “TransferData” functions to transfer cell type labels from a reference dataset onto a new query dataset. For the “FindTransferAnchors” function, we use

the default parameters, *i.e.*, `reduction = "pcaproject"`, `npcs = 30`, `dims = 1:30`, `k.anchor = 5`, `k.filter = 200`, `k.score = 30`, `max.features = 200`, `nn.method = "rann"`, `normalization.method = "LogNormalize"`, `approx.pca = TRUE`. For the “TransferData” function, we also use the default parameters, *i.e.*, `weight.reduction = "pcaproject"`, `dims = 1:30`, `k.weight = 50`, `sd.weight = 1`.

According to its vignette, Seurat v3 provides the options of using PCA as the default way (PCA-based Seurat v3) and CCA as an alternative (CCA-based Seurat v3) to project the structure of a reference onto the query, which are used in the anchor weighting and label transfer steps¹². To investigate whether the CCA-based Seurat v3 gains better performance than PCA-based Seurat v3, we apply the CCA-based Seurat v3 to the 26 datasets from three scenarios that have true labels (Supplementary Fig. 14a). We also compare the performance of CCA-based Seurat v3 versus scGCN accordingly (Supplementary Fig. 14b). For evaluation, we use the accuracy score (Acc) to evaluate the performance of the CCA-based Seurat. In Supplementary Fig. 14, different colors represent different paired reference-query datasets, while different symbols represent different label transfer scenarios.

a
△ Within-data

- △ GSE108989
- △ GSE115746
- △ GSE118389
- △ GSE72056
- △ GSE98638
- △ GSE99254
- △ GSM3271044
- △ GSM3271045
- △ phs001790
- △ SRP073767

● Cross-platform

- E-MTAB-5061 – GSE84133
- GSE81608 – GSE84133
- GSE84133 – E-MTAB-5061
- GSE84133 – GSE85241
- GSE85241 – GSE84133
- MCA 10x – MCA Smart-Seq2
- PBMC Cel-seq – PBMC 10xV3
- PBMC Cel-seq – PBMC InDrop
- PBMC DropSeq – PBMC Smart-Seq2
- PBMC DropSeq – PBMC10xV3
- PBMC InDrop – PBMC Cel-Seq
- PBMC Seq-well – PBMC Smart-Seq2

□ Cross-species

- GSE84133: human – mouse
- GSE84133: mouse – human
- phs001790 (human) – GSE115746 (mouse)
- GSE115746 (mouse) – phs001790 (human)

b
Supplementary Fig. 14 Performance of CCA-based Seurat v3, PCA-based Seurat v3, and scGCN. a Performance of CCA-based Seurat v3 versus PCA-based Seurat v3. **b** Performance of CCA-based Seurat v3 versus scGCN. The dashed line represents equivalent accuracy between two methods. Different symbols represent different learning scenarios, i.e., within-data, cross-platform, and cross-species. Different colors represent different paired reference-query datasets.

Supplementary Fig. 14a shows that PCA-based Seurat v3 has better performance on 20 of the total 26 datasets, except for 6 paired data including 1 reference-query pair from within-data, 2 from cross-platform, and 3 from cross-species scenario. Specifically, regarding the 4 reference-query data pairs from within-data (GSE118389), cross-platform datasets (MCA 10x – MCA Smart-Seq2, GSE84133 – GSE85241), and cross-species datasets (GSE84133: mouse – human), CCA-based Seurat shows limited increase of accuracy (0.0093%, 7.24%, 1.30%, and 3.16%).

Interestingly, CCA-based Seurat shows higher accuracy (17.99% and 15.03% more than PCA-based Seurat) at the other two cross-species datasets, *i.e.*, phs001790 (human) – GSE115746 (mouse) and phs001790 (mouse) – GSE115746 (human). Across all 26 datasets, the mean accuracy of CCA-based Seurat is 78.96%, which is lower than PCA-based Seurat (mean Acc = 82.18%). These results indicate that PCA-based Seurat v3 performs better than CCA-based Seurat v3 in most cases. The observed better results of PCA-based Seurat v3 may be because that it uses not only PCA in dimension reduction to identify individual sources of variation, but also CCA in constructing anchors to identify shared sources of variation across datasets. In contrast, CCA-based Seurat v3 uses redundant shared information that ignore individual variation of each dataset.

To further demonstrate our scGCN’s performance, we compare it with CCA-based Seurat v3 in Supplementary Fig. 14b. We find that scGCN has better performance on all the 26 datasets. The mean accuracy of scGCN across all datasets is 88.57%, which is better than CCA-based Seurat v3 (mean Acc = 78.96%). Specifically, for the within-data “GSE108989”, scGCN outperforms CCA-based Seurat v3 the most with 27.9% more accuracy. For the cross-platform dataset “MCA 10x – MCA Smart-Seq2”, scGCN has 15.93% higher accuracy. Overall, these comparisons demonstrate the superior performance of scGCN than CCA-based Seurat v3. We add the related revisions in the updated manuscript Page 11 and the Supplementary Note 7.

3. The authors didn’t introduce how to deal with unknown cell types in query datasets. The measures have ignored new type cells not appearing in the reference cells, will they be assigned to known labels or identified as new types? If wrongly assigned, this may cause problems in many situations. It’s necessary to expand the experiments about the accuracy of unknown cell types.

Response: Thanks for the reviewer’s comments. We provide comprehensive experiments to evaluate our method for identifying unknown cells in query data.

Statistical metrics

To identify potential unknown cells in query data, we provide a screening step in our scGCN model using two statistical metrics, entropy score and enrichment score, representing mixture and enrichment. Specifically, all cells in query data are grouped to different clusters by modularity-based community detection¹⁷. For each query cluster, we measure its mixture and enrichment based on the inter-data graph of scGCN. Our rationale is that, a query cluster of unknown cell type is more likely to have random links to different cell types in the reference data, while a query cluster of known cell type is more likely to link to a specific cell type in the reference data. In this way, unknown cells can be identified by the two statistical metrics. Mathematical definitions are illustrated below.

Entropy score: For a cluster h in the query data, the mixture of this cluster is defined by the information entropy of normalized enrichment scores. That is,

$$H_h = - \sum_c^C \frac{S_{c,h}}{\sum_c^C S_{c,h}} \log \frac{S_{c,h}}{\sum_c^C S_{c,h}} \quad \text{and} \quad S_{c,h} = \frac{m_{c,h} / \sum_c^C m_{c,h}}{n_c / \sum_c^C n_c}$$

, where c is a specific cell type, C is the set of all cell types in reference data, $m_{c,h}$ is the number of cells in query cluster h that are linked to cell type c in reference data by the inter-data graph of scGCN, and n_c is the number of cells belonging to cell type c in reference data.

Enrichment score: For a cluster h in the query data, the enrichment score ES_h is defined by the normalized enrichment of the most enriched cell type. That is,

$$ES_h = \frac{\max_{c \in C} S_{c,h}}{\sum_c S_{c,h}}$$

In summary, for a cluster in query data, the entropy score describes whether this cluster in query data is dominated by a specific cell type, and the enrichment score describes how strong this cell type is enriched. Thus, if the query cell cluster h has higher entropy and lower enrichment, these cells should be assigned as unknown cell types.

Performance evaluation

To demonstrate the performance of the two statistical metrics in identifying cells of unknown types, we perform a comprehensive evaluation experiment based on the cross-platform data and cross-species data used in Fig. 3 and Fig. 4. We construct evaluation datasets and evaluate the performance using the area under the receiver operating characteristics curves (AUC). Specifically, for a given reference-query data pair with l cell types in reference data, totally l evaluation datasets are derived, each with a specific cell type removed from the reference and thus the corresponding cells in query data are unknown. Both entropy score and enrichment score are calculated in each of the l evaluation datasets, and the overall performance of detecting unknown cells is summarized into AUC scores of the two metrics, respectively.

Based on the 12 reference-query pairs from different platforms and 4 pairs from different species, the performance of recognizing unknown cells is shown in the receiver operating characteristics (ROC) curves with calculated AUC values (Supplementary Fig. 13). For the paired datasets from different platforms, the ROC curves of entropy and enrichment scores are shown in Supplementary Fig. 13a-b. The average AUCs of entropy score and enrichment score are 0.857 and 0.814, respectively. Specifically, the entropy and enrichment scores both show high AUC for the GSE84133 – E-MTAB-5061 datasets. Similarly, for the paired datasets from different species, we identify the ROC curves of entropy and enrichment score in Supplementary Fig. 13c-d. The average AUCs of entropy score and enrichment score are 0.805 and 0.785, respectively. These two scores are shown with highest AUC for the GSE84133: human – mouse dataset. As the reference-query pair is from the same dataset that doesn't have unknown cell types, we do not consider the within-data scenario. Given the above evaluations, we demonstrate that the two statistical metrics in scGCN are effective in identifying unknown cells in query datasets. The related revisions can be found in the updated manuscript Page 10 and the Supplementary Note 6.

Supplementary Fig. 13 Performance of entropy score and enrichment score in identifying unknown cells in query data. ROC curves of entropy score (a) and enrichment score (b) on the 12 pairs of reference-query data from different platforms. ROC curves of entropy score (c) and enrichment score (d) on the 4 pairs of reference-query data

from different species. Different colors represent different reference-query pairs. The corresponding AUC score is indicated within the parenthesis.

4. They authors didn't describe how to select the reference datasets? The performance of methods are known to highly depend on datasets, and thus the comparisons may be misleading if manually picking datasets from the large number of single cell datasets publicly available.

Response: Thanks for the reviewer's comments. We agree with the reviewer that the comparisons should be independent to the selection of the reference datasets. Therefore, we systematically examine the performance of scGCN in two aspects: 1). Performance comparisons are independent of reference data selection (Supplementary Fig. 6); 2). Systematic evaluation using benchmarking data (Supplementary Fig. 7). As Seurat v3 performs better than Conos, scmap, and CHETAH, we only show the comparisons between scGCN and Seurat v3.

1). Performance comparisons are independent of reference data selection

Here we examine whether the outperformance of scGCN depends on the selection of reference data in the reference-query data pairs. For each reference-query pair shown in our manuscript, we reverse the pair and evaluate the performance. That is, the data previously used as reference is used as query, and vice versa. We do not evaluate the within-data scenario since the reference and query data are already randomly selected.

a**b**
Supplementary Fig. 6 Performance of scGCN on original and reversed reference-query data pairs. a Performance of scGCN versus Seurat v3 on the 11 unique pairs of reference-query data from Fig. 3. **b** Performance of scGCN versus Seurat v3 on the 11 reversed pairs of reference-query data. The dashed line represents equivalent accuracy between two methods. Different colored symbols represent different reference-query pairs.

For the cross-platform (Fig. 3) and cross-species (Fig. 4) scenarios, there are a total of 11 unique combinations of reference and query data. The performance of these unique combinations is shown in Supplementary Fig. 6a, and the results of reversed pairs are shown in Supplementary

Fig. 6b. In Supplementary Fig. 6a, as shown in our manuscript, scGCN has better performance with average accuracy of 86.6% than Seurat v3 (mean Acc = 78.6%). For these reversed pairs (Supplementary Fig. 6b), the average accuracy of scGCN is 86.9% that still outperforms Seurat v3 (mean Acc = 82.1%). These comparisons show the superior performance of scGCN is independent of our reference data selection.

2). Systematic evaluation using benchmarking data

We also agree with the reviewer that our selected datasets may not be representative. Therefore, we use a well-recognized benchmarking collection of single-cell datasets covering 13 major platforms¹⁸ and completely examine all possible reference-query combinations, which allows unbiased evaluation of model performance.

Supplementary Fig. 7 Performance of scGCN on comprehensive benchmarking datasets. Performance of scGCN versus Seurat v3 on 156 reference-query pairs from 13 different platforms, including C1HT-medium, MARS-Seq, Quartz-Seq2, SCRBS-Seq, Smart-Seq2, C1HT-small, CEL-Seq2, Chromium, Chromium (sn), ddSEQ, Drop-Seq, ICCELL8, and inDrop. The dashed line represents equivalent accuracy between two methods. Different colors represent reference-query pairs from different platforms.

As shown in Supplementary Fig. 7, we randomly pick one dataset from the 13 platforms as the reference and the other one as the query data. In this way, we get 156 different combinations of reference and query datasets from these platforms. The accuracy scores of scGCN versus Seurat v3 on all these 156 pairs are shown accordingly. Across all the 156 pairs, scGCN shows an average accuracy at 76.27% while Seurat v3 achieves the average accuracy at 71.12%. Specifically, scGCN shows better accuracy than Seurat v3 on 121 paired datasets with an average 7.28% increase of accuracy. On the other 34 paired datasets, Seurat v3 shows slightly higher accuracy with an average 2.30% increase of accuracy. scGCN's top outperformed cases are ICCELL8 – inDrop, inDrop – CEL-Seq2, and MARS-Seq – Smart-Seq2, with increased accuracy of 33.91%, 32.21%, and 31.73%, respectively.

Altogether, with the comprehensive comparisons on reversed reference-query pairs and a collection of benchmarking data, scGCN consistently outperforms Seurat v3, which is not due to

manually picked datasets. The related revisions are shown in the updated manuscript Page 6 and the Supplementary Note 2.

5. With the development of single-cell sequencing technology, it is worth describing computational issues e.g. memory usage and training time for the method, especially in large-scale data sets.

Response: Thanks for the reviewer's comments. We demonstrate the scalability of the scGCN method in large-scale datasets. Specifically, the memory usage and the computing time are profiled with respect to the sample size, from 100k up to 1 million cells, using the dataset generated by Cao J, et. al.¹⁹. The computational costs of scGCN are compared with Seurat v3 on a computer with 64 GB memory and 3.6 GHz Intel Core i9 processor. To enable label transfer for million-level or larger datasets, a batched approach is used for both methods. Briefly, the original dataset is split into batches for computing and the outputs from each batch are summarized into the final results. We show the computational time and average memory usage for different sample sizes (*i.e.*, 100k, 200k, ..., 1million) in Supplementary Fig. 8.

As shown in Supplementary Fig. 8, scGCN and Seurat v3 are comparable in computational time and memory usage. Specifically, in Supplementary Fig. 8a, the computing time of scGCN is linearly scalable with respect to the increase of sample size, ranging from 0.88 hours for 100k cells to 9.08 hours for 1 million cells, which are of the same magnitude with Seurat v3 that ranges from 0.48 hours to 5.11 hours respectively. Besides, we also investigate the batch-wise average memory usage of scGCN comparing with Seurat v3 (Supplementary Fig. 8b). scGCN shows lower memory usage (median: 2.11GB, IQR: 2.09 to 2.14GB) than Seurat v3 (median: 2.17GB, IQR: 2.14 to 2.19GB). Overall, scGCN is efficient and scalable for large-size single-cell dataset. We have revised the manuscript accordingly in Page 6 and the Supplementary Note 3.

a**b**
Supplementary Fig. 8 Computational performance of scGCN on large-size single-cell dataset. **a** Computation time of scGCN and Seurat on datasets of different sample sizes. The x-axis represents different sample sizes, the y-axis represents the corresponding running time in hours. **b** Batch-wise average memory usage of scGCN and Seurat across batches. The x-axis represents the memory usage (GB) and the y-axis represents the counts of batches, with each batch consisting of 5,000 cells.

References

- 1 Hie, B., Bryson, B. & Berger, B. Efficient integration of heterogeneous single-cell transcriptomes using Scanorama. *Nature Biotechnology* **37**, 685-691, doi:10.1038/s41587-019-0113-3 (2019).
- 2 Kiselev, V. Y., Yiu, A. & Hemberg, M. scmap: projection of single-cell RNA-seq data across data sets. *Nature Methods* **15**, 359-362, doi:10.1038/nmeth.4644 (2018).
- 3 DePasquale, E. A. K. *et al.* cellHarmony: cell-level matching and holistic comparison of single-cell transcriptomes. *Nucleic Acids Research* **47**, e138-e138, doi:10.1093/nar/gkz789 (2019).
- 4 Maaten, L. v. d. & Hinton, G. Visualizing data using t-SNE. *Journal of machine learning research* **9**, 2579-2605 (2008).
- 5 McInnes, L., Healy, J. & Melville, J. Umap: Uniform manifold approximation and projection for dimension reduction. *arXiv preprint arXiv:1802.03426* (2018).
- 6 Moon, K. R. *et al.* Visualizing structure and transitions in high-dimensional biological data. *Nat Biotechnol* **37**, 1482-1492, doi:10.1038/s41587-019-0336-3 (2019).
- 7 Veličković, P. *et al.* Graph attention networks. *arXiv preprint arXiv:1710.10903* (2017).
- 8 Hamilton, W., Ying, Z. & Leskovec, J. in *Advances in neural information processing systems*. 1024-1034.
- 9 Defferrard, M., Bresson, X. & Vandergheynst, P. Convolutional neural networks on graphs with fast localized spectral filtering. *Advances in neural information processing systems* **29**, 3844-3852 (2016).
- 10 Chami, I., Ying, Z., Ré, C. & Leskovec, J. in *Advances in neural information processing systems*. 4868-4879.
- 11 Xu, B., Shen, H., Cao, Q., Qiu, Y. & Cheng, X. Graph wavelet neural network. *arXiv preprint arXiv:1904.07785* (2019).
- 12 Stuart, T. *et al.* Comprehensive integration of single-cell data. *Cell* **177**, 1888-1902. e1821 (2019).
- 13 Wen, W. *et al.* Immune cell profiling of COVID-19 patients in the recovery stage by single-cell sequencing. *Cell Discovery* **6**, 31, doi:10.1038/s41421-020-0168-9 (2020).
- 14 Stuart, T. & Satija, R. Integrative single-cell analysis. *Nature Reviews Genetics* **20**, 257-272, doi:10.1038/s41576-019-0093-7 (2019).
- 15 Butler, A., Hoffman, P., Smibert, P., Papalexi, E. & Satija, R. Integrating single-cell transcriptomic data across different conditions, technologies, and species. *Nature Biotechnology* **36**, 411-420, doi:10.1038/nbt.4096 (2018).
- 16 Haghverdi, L., Lun, A. T. L., Morgan, M. D. & Marioni, J. C. Batch effects in single-cell RNA-sequencing data are corrected by matching mutual nearest neighbors. *Nature biotechnology* **36**, 421-427, doi:10.1038/nbt.4091 (2018).
- 17 Waltman, L. & Van Eck, N. J. A smart local moving algorithm for large-scale modularity-based community detection. *The European physical journal B* **86**, 1-14 (2013).
- 18 Mereu, E. *et al.* Benchmarking single-cell RNA-sequencing protocols for cell atlas projects. *Nat Biotechnol* **38**, 747-755, doi:10.1038/s41587-020-0469-4 (2020).
- 19 Cao, J. *et al.* A human cell atlas of fetal gene expression. *Science* **370** (2020).

REVIEWER COMMENTS

Reviewer #1 (Remarks to the Author):

The authors have answered all my comments with thorough evaluations that form an extensive ablation study. I have no further comments other than recommending to accept this manuscript for publication.

Reviewer #2 (Remarks to the Author):

Most of my questions have been addressed.

The authors use a batched approach for scGCN and Seurat v3 to split million-level or larger datasets, and run the experiments on a 64-GB-memory computer with 9.08 hours. However, in the batch-correction benchmark paper [1], Seurat has very high memory usage (~1000GB) for a mouse brain dataset with 833,206 cells. Can the authors give more details about the batch-splitting approach for scGCN and Seurat v3? Will the final accuracy be affected by batch-correction and model training on batch-splitting data independently?

Reference

[1] Tran, H.T.N., Ang, K.S., Chevrier, M. et al. A benchmark of batch-effect correction methods for single-cell RNA sequencing data. *Genome Biol* 21, 12 (2020). <https://doi.org/10.1186/s13059-019-1850-9>

Reviewers' Comments to Author:

REVIEWER COMMENTS

Reviewer #1 (Remarks to the Author):

The authors have answered all my comments with thorough evaluations that form an extensive ablation study. I have no further comments other than recommending to accept this manuscript for publication.

Reviewer #2 (Remarks to the Author):

Most of my questions have been addressed.

The authors use a batched approach for scGCN and Seurat v3 to split million-level or larger datasets, and run the experiments on a 64-GB-memory computer with 9.08 hours. However, in the batch-correction benchmark paper [1], Seurat has very high memory usage (~1000GB) for a mouse brain dataset with 833,206 cells. Can the authors give more details about the batch-splitting approach for scGCN and Seurat v3? Will the final accuracy be affected by batch-correction and model training on batch-splitting data independently?

Reference

[1] Tran, H.T.N., Ang, K.S., Chevrier, M. et al. A benchmark of batch-effect correction methods for single-cell RNA sequencing data. *Genome Biol* 21, 12 (2020). <https://doi.org/10.1186/s13059-019-1850-9> [doi.org]

Response: Thanks for the reviewer's comments.

Regarding the batch-splitting strategy, we first construct reference and query datasets using Cao J, et. al.¹ that consists of 77 cell types and 4,062,980 cells, then split the query dataset into batches and independently learn cell labels for each batch, and finally summarize the results from each batch as the learned cell labels for the whole query dataset. 1) Data preparation. From Cao's original dataset, each cell type with more than 500 cells are retained for evaluation, thus resulting in 4,060,392 cells of 64 cell types. From these 4,060,392 cells, we randomly pick 200 cells for each of the 64 cell types and combine them as the reference data. From the rest of cells, we randomly sample cells as the query data. 2) Batch-splitting. The query data are then randomly split into batches, with each batch consisting of 5,000 cells. For example, if 1 million cells are randomly selected as the query data, totally 200 batches will be constructed. 3) Batch-splitting learning. For each batch of the query data, we use both scGCN and Seurat v3 to identify its transferred labels based on the reference data. 4) Summarization. The identified cell types for each batch are then summarized together to calculate the accuracy of the whole query data.

We next comprehensively investigate the accuracy of the batch-splitting approach comparing with the overall-learning approach for both scGCN and Seurat v3 at different query sample size scales, from 100k cells to 1 million cells. Here the "overall-learning" approach denotes using scGCN or Seurat v3 without batch-splitting. To present robust comparisons, for each query sample size, we

repeat the whole experiment for 4 times, each with a new randomly sampled data. The label transfer accuracy is shown in Supplementary Fig. 9a. Different colors represent different strategies of scGCN and Seurat v3. scGCN's overall-learning ($87.89\% \pm 0.23\%$) has slightly higher accuracy than its batch-splitting ($85.93\% \pm 0.11\%$) across all query sample sizes. In contrast, for Seurat v3, the accuracy of the overall-learning approach ($63.46\% \pm 3.48\%$) is much lower than the batch-splitting approach ($76.85\% \pm 0.15\%$). In all cases, scGCN outperforms Seurat v3. Our analysis suggests that scGCN batch-splitting approach demonstrates comparable performance with scGCN overall-learning approach. Considering the superior scalability of the batch-splitting approach, in practice, we recommend using the batch-splitting approach when implementing scGCN on large query data.

To gain further insights into the scalability of the batch-splitting versus overall-learning, in Supplementary Fig. 9b, we apply both scGCN and Seurat v3 on query data ranging from 0.6k to 1 million cells using one of the four sampled datasets. As shown in the figure, scGCN and Seurat v3 demonstrate similar performance when cells are less than 2.5k, but they start to diverge on larger query datasets. Notably, when the query sample size is small (0.6k to 5k cells), the accuracies of both overall-learning and batch-splitting approaches of scGCN improve when the query sample size grows. When the query sample size is beyond 5k cells, both approaches reach maximum and stable performance. In contrast, though the Seurat v3's batch-splitting presents with small variations across different number of cells, the accuracy of the Seurat v3's overall-learning approach gradually decreases when query data size is over 10k. This also explains why Seurat v3 shows lower accuracy of overall-learning in large-size dataset in Supplementary Fig. 9a.

The superior performance of scGCN over Seurat v3 on large query data may be due to that, as a semi-supervised approach, scGCN effectively learns the topological information from both reference and query data. A larger query data provides more topological information that benefits scGCN, and such benefit of a larger query data saturates at 5k cells. The performance of the Seurat v3 overall-learning approach deteriorates when the query data is over 10k cells, which might be due to that the large query data dominates the learning and dilutes the information from the reference data.

Overall, our results suggest that, in transfer-learning scenario, batch-splitting and overall-learning demonstrate similar accuracy when implementing scGCN on large query data. Therefore, scGCN is efficient and scalable for large-size single-cell dataset with high accuracy. We have revised the manuscript and cited Tran et.al² accordingly on Page 6 and in Supplementary Note 3. All detailed accuracy is deposited in the source data file.

a

b

Supplementary Fig. 9 Performance of overall-learning versus batch-splitting on large-size single-cell dataset. a Accuracy of scGCN and Seurat v3 on 4 randomly sampled datasets of different cell numbers using batch-splitting and

overall-learning, respectively. **b** Accuracy of scGCN and Seurat v3 on sample sizes ranging from 0.6k to 1 million. X-axis represents cases of different samples sizes. Solid lines represent overall-learning results and dashed lines represent batch-splitting results. Different colors represent different methods.

References

- 1 Cao, J. *et al.* A human cell atlas of fetal gene expression. *Science* **370** (2020).
- 2 Tran, H. T. N. *et al.* A benchmark of batch-effect correction methods for single-cell RNA sequencing data. *Genome biology* **21**, 1-32 (2020).

REVIEWERS' COMMENTS

Reviewer #2 (Remarks to the Author):

I have no further comments.

REVIEWERS' COMMENTS

Reviewer #2 (Remarks to the Author):

I have no further comments.

Response: Thanks for the reviewer's comments. We are pleased that we have addressed your concerns.